# UNCERTAINTY ESTIMATION
# VIA HYPERSPHERICAL CONFIDENCE MAPPING

**Eunseo Choi**
KAIST
ces302@kaist.ac.kr

**Ho-Yeon Kim**
Samsung Electronic Co., Ltd
hoyn.kim@samsung.com

**Jaewon Lee**
Samsung Electronic Co., Ltd
jw0928.lee@samsung.com

**Taeyong Jo**
Samsung Electronic Co., Ltd
ty.jo@samsung.com

**Myungjun Lee**
Samsung Electronic Co., Ltd
myung01.lee@samsung.com

**Heejin Ahn**
KAIST
heejin.ahn@kaist.ac.kr

## ABSTRACT

Quantifying uncertainty in neural network predictions is essential for high-stakes domains such as autonomous driving, healthcare, and manufacturing. While existing approaches often depend on costly sampling or restrictive distributional assumptions, we propose **Hyperspherical Confidence Mapping (HCM)**, a simple yet principled framework for sampling-free and distribution-free uncertainty estimation. HCM decomposes outputs into a magnitude and a normalized direction vector constrained to lie on the unit hypersphere, enabling a novel interpretation of uncertainty as the degree of violation of this geometric constraint. This yields deterministic and interpretable estimates applicable to both regression and classification. Experiments across diverse benchmarks and real-world industrial tasks demonstrate that HCM matches or surpasses ensemble and evidential approaches, with far lower inference cost and stronger confidence–error alignment. Our results highlight the power of geometric structure in uncertainty estimation and position HCM as a versatile alternative to conventional techniques.

## 1 INTRODUCTION

Neural networks have achieved remarkable success across a wide range of domains such as computer vision, natural language processing, and scientific modeling. As these models are increasingly deployed in high-stakes applications, including autonomous driving, medical diagnosis, and semiconductor manufacturing, the reliability of their predictions becomes critical (Kendall & Gal, 2017; Lakshminarayanan et al., 2017; Esteva et al., 2019). In safety-critical settings, erroneous predictions can lead to severe real-world consequences. Thus, quantifying the uncertainty associated with neural network outputs is not merely desirable but essential for trustworthy AI (Gneiting et al., 2007; Guo et al., 2017).

A widely adopted strategy for uncertainty estimation is the use of *sampling-based* methods, such as Monte Carlo (MC) dropout (Gal & Ghahramani, 2016) and Deep Ensembles (Lakshminarayanan et al., 2017). These methods approximate predictive uncertainty by performing multiple stochastic forward passes or aggregating outputs from several independently trained models. While often effective, they incur substantial computational and memory overhead, limiting their practicality in real-time or resource-constrained scenarios (Ahmed et al., 2024).

To avoid repeated sampling, *distribution-based* methods directly model predictive uncertainty through parametric families such as Gaussian regression (Kendall & Gal, 2017), Dirichlet-based classification (Malinin & Gales, 2018), and evidential learning (Amini et al., 2020). These ap-

Table 1: Comparison of uncertainty estimation methods

| Method | Sampling-free | Dist-free | Real-time | Task-agnostic | Interpretable |
|---|---|---|---|---|---|
| Sampling | ✗ | ✓ | ✗ | ✓ | ✗ |
| Distribution | ✓ | ✗ | ✓ | ✓ | – |
| Interval | ✓ | ✗ | ✓ | ✓ | – |
| Similarity | ✓ | ✓ | ✓ | ✗ | ✓ |
| **HCM (ours)** | ✓ | ✓ | ✓ | ✓ | ✓ |

proaches provide deterministic inference but rely on strong distributional priors (e.g., Gaussian or Dirichlet) that may fail to capture multimodal or complex forms of uncertainty.

In contrast, *interval-based* methods construct prediction intervals that contain the true value with a prescribed probability. Quantile Regression (Koenker, 2005) and Lower Upper Bound Estimation (Khosravi et al., 2011) estimate interval bounds without explicit distributional assumptions, while conformal prediction (Vovk et al., 2005; Shafer & Vovk, 2008) ensures statistically valid coverage for arbitrary models. However, these methods often require multiple quantile outputs, carefully designed objectives, and typically guarantee only marginal rather than per-sample coverage.

Finally, *similarity-based* methods estimate uncertainty from feature-space similarity or density. Examples include distance-based approaches (Van Amersfoort et al., 2020; Mukhoti et al., 2023), energy-based approaches (Liu et al., 2020), and recent similarity-driven uncertainty methods proposed in text classification (Gong et al., 2022; Kotelevskii et al., 2022; Vazhentsev et al., 2023). Although effective for classification, these approaches rely on class prototypes or density estimation and therefore do not extend naturally to regression.

In this work, we introduce a new uncertainty estimation framework called **Hyperspherical Confidence Mapping (HCM)**. Our method avoids both sampling and distributional assumptions while maintaining high interpretability, computational efficiency, and task-agnostic applicability. HCM decomposes the model output into two components: a magnitude term $R$, defined as the output norm, and a direction vector $d$, obtained by normalizing with $R$ so that $\|d\| = 1$. This decomposition is not a mere heuristic, but a reformulation of the original unconstrained prediction problem into a constrained optimization problem under the unit-norm constraint on $d$.

We interpret the violation of this hyperspherical constraint — deviations from the unit norm — as a measure of predictive uncertainty. This geometric interpretation naturally integrates into end-to-end training, enabling a confidence representation grounded in constraint satisfaction. By directly exploiting this formulation, HCM provides deterministic and interpretable uncertainty estimates without relying on distributional priors and repeated sampling. As summarized in Table 1, HCM uniquely combines sampling-free, distribution-free, task-agnostic, and real-time properties in a single framework, making it applicable to both regression and classification and scalable to real-world deployment. To facilitate reproducibility, we make the source code for our CIFAR-10, Two-Moons, and 1D regression experiments publicly available at https://github.com/Abandoned-Puppy/HCM.

**Our contributions.**

- We propose **Hyperspherical Confidence Mapping (HCM)**, a novel uncertainty estimation framework that is sampling-free, distribution-free, and task-agnostic.

- We establish theoretical foundations by reformulating prediction as a constrained optimization problem and interpreting constraint violations as uncertainty, which is provably related to prediction error.

- We validate HCM extensively on classification and regression datasets, demonstrating superior calibration, confidence–error alignment, and competitive out-of-distribution detection compared to existing baselines.

- Beyond benchmarks, we showcase the applicability of HCM in *real-world semiconductor manufacturing tasks*, highlighting its scalability and practical impact.

## 2 METHOD

In this section, we present the details of our proposed method and demonstrate that our geometric formulation provides reliable and interpretable estimates of prediction error.

### 2.1 PROBLEM FORMULATION

We denote the input by $x \in \mathcal{X} \subseteq \mathbb{R}^{D^*}$ and the corresponding target by $y \in \mathbb{R}^D$. For classification with $C$ classes, we set $D = C$, representing the target as a one-hot vector in $\mathbb{R}^D$. The network output $f_\theta(x) \in \mathbb{R}^D$ is trained to regress toward this target, and the predicted class is obtained via $\arg\max f_\theta(x)$. Thus both regression and classification are treated within the same regression framework in $\mathbb{R}^D$. The goal is not only to produce accurate predictions $\hat{y} = f_\theta(x)$, but also to quantify the associated uncertainty in a deterministic and efficient way.

### 2.2 HYPERSPHERICAL DECOMPOSITION

We propose **Hyperspherical Decomposition**, a novel approach that separates a model's output into a scaler **magnitude** $R$ and a **unit-norm direction vector** $d$. Given a target $y \in \mathbb{R}^D$, we reformulate it as $y = Rd$, where $R \in \mathbb{R}^+$, $d \in \mathbb{R}^D$, and $\|d\|_2 = 1$. Our model is trained to predict these two components, $\hat{R}$ and $\hat{d}$, to form the final prediction $\hat{y} = \hat{R}\hat{d}$. This decomposition is not a mere heuristic but a principled way to impose a geometric constraint that is both **unbiased** and **prior-free**. While other constraints can break the inherent symmetry between output dimensions and inject directional biases, our hyperspherical decomposition ensures that all dimensions are treated equally, leading to a geometrically consistent representation. For scalar regression ($D = 1$), we embed the target into a minimal $D = 2$ space via duplication, $y_{\exp} = (y, y)$, enabling the same decomposition without modifying the core formulation.

This formulation is distinct from prior hyperspherical approaches for text calibration, such as (Gong et al., 2022), which apply a hyperspherical constraint only to the classification logits. In contrast, we decompose the *target* itself into a magnitude–direction pair and derive uncertainty from violations of this geometric constraint, enabling a unified treatment of both regression and classification.

### 2.3 UNCERTAINTY AND CONSTRAINT VIOLATION

In machine learning, prediction problems are typically formulated as an unconstrained optimization problem. With our decomposition, we reformulate it as a constrained optimization problem

$$\min_{\hat{R},\hat{d}} \mathcal{L}(\hat{R}\hat{d}, y) \quad \text{subject to } \|\hat{d}\|_2 = 1. \tag{1}$$

This is the *primal* formulation, where the direction vector $\hat{d}$ is required to lie on the unit hypersphere. In practice, we do not enforce this condition as a hard constraint. Since the ground-truth $d$ satisfies $\|d\|_2 = 1$, the training objective inherently drives $\hat{d}$ toward the unit norm as it learns to approximate $d$. Thus, $\hat{d}$ may still deviate slightly from the hyperspherical surface, and such deviations serve as informative signals of prediction reliability. We define the uncertainty score as

$$u(x) := \hat{R}(x)\big|\|\hat{d}(x)\|_2 - 1\big|, \tag{2}$$

which directly measures the degree of constraint violation in the output direction. This score is computed deterministically from the model output without requiring sampling, labels, or auxiliary networks, making it efficient and lightweight.

**Notation.** Throughout this section, let the ground truth be $y = Rd$ with $\|d\|_2 = 1$, and the prediction be $\hat{y} = \hat{R}\hat{d}$. Define the errors $e_y := \hat{y} - y$, $e_d := \hat{d} - d$, and $e_R := \hat{R} - R$. For simplicity, we write $\hat{R}$ and $\hat{d}$ when the input $x$ is clear from context, and use $\hat{R}(x)$ and $\hat{d}(x)$ when we need to make the dependence explicit.

### 2.4 TRAINING OBJECTIVE

We minimize the total loss

$$\mathcal{L}_{\text{total}} = \phi_d\big(R\|e_d\|_2\big) + \phi_R(e_R) + \lambda_{\text{norm}}\, \phi_{\text{norm}}\big(\big|\|\hat{d}\|_2 - 1\big|\big). \tag{3}$$

The first two terms supervise the direction and magnitude, respectively, while the last term softly enforces the unit-norm constraint with weight $\lambda_{\text{norm}}$. Here, each $\phi_\star(\cdot)$ denotes a member of the same loss family (e.g., power-$p$, Huber, or smooth-$\ell_1$), providing a unified formulation that generalizes beyond the squared loss. This design allows the loss to flexibly adapt its curvature or robustness while maintaining consistent treatment across the direction, magnitude, and norm components.

**Connection with the primal problem.**  This objective can be viewed as a soft relaxation of the constrained formulation in (1), with the norm penalty acting as the regularizer. Compared to the exact expansion of (1), which is given in Appendix A.3, it simplifies the coupling between the magnitude $\hat{R}$ and the direction $\hat{d}$, resulting in more stable optimization.

## 2.5 THEORETICAL MOTIVATION

The hyperspherical decomposition induces a geometric consistency constraint $\|\hat{d}(x)\|_2 \approx 1$, and violations of this constraint provide information about how reliably the model predicts a given input. Rather than attempting to explicitly disentangle epistemic and aleatoric uncertainty, we interpret the uncertainty score

$$u(x) := \hat{R}(x) \left| \|\hat{d}(x)\|_2 - 1 \right|$$

as a deterministic indicator of prediction reliability whose behavior reflects both data scarcity and intrinsic noise.

**Lower bound on prediction error.**  We first establish a basic relationship between $u(x)$ and the prediction error.

**Proposition 1** (Lower bound on prediction error). *Let $\epsilon := \frac{|e_R|}{\hat{R}\|e_d\|_2}$ with $\hat{R}\|e_d\|_2 > 0$. Then*

$$\|e_y\|_2 \geq u(x)\,(1 - \epsilon). \tag{4}$$

*When $\epsilon \ll 1$, the score $u(x)$ acts as a prediction-dependent surrogate for the true error.*

*Proof.* Since $e_y = \hat{R}e_d + e_R d$, the triangle and reverse-triangle inequalities give

$$\hat{R}\|e_d\|_2 - |e_R| \ \leq \ \|e_y\|_2 \ \leq \ \hat{R}\|e_d\|_2 + |e_R|.$$

Because $\hat{R}\|e_d\|_2 \geq \hat{R}|\|\hat{d}\|_2 - \|d\|_2| = u(x)$, the claim follows. $\qquad\square$

**Remark 1.** *In typical trained models, the term $|e_R|$ is substantially smaller than $\hat{R}\|e_d\|_2$, so the quantity $u(x)$ serves as a reliable indicator of large prediction error: when $u(x)$ is large, the error $\|e_y\|_2$ must also be large. Thus, $u(x)$ acts as a deterministic lower-bound indicator of prediction error, providing a scale-consistent measure of prediction reliability.*

This direct and monotonic relationship between $u(x)$ and the minimum achievable error makes the score inherently interpretable: its magnitude has a clear meaning—larger values certify that accurate prediction is mathematically unattainable.

**Scale-aware variability under additive noise.**  To examine how the geometric structure responds to input-dependent noise, we also consider a variance-like quantity

$$\hat{\sigma}^2(x) := \tfrac{1}{D-1}\, u(x)\left(\hat{R}(x)(1 + \|\hat{d}(x)\|_2)\right),$$

which combines the lower-bound factor $u(x)$ with an approximate upper-bound factor derived from the predicted magnitude. Although $\hat{\sigma}(x)$ is not a probabilistic variance, it increases predictably with noise level and provides a deterministic measure of aleatoric variability.

**Proposition 2** (Behavior under Gaussian noise). *If $y = g(x) + \xi$ with zero-mean Gaussian noise of variance $\sigma^2$, then*

$$\hat{\sigma}^2(x) = \sigma^2 + O\!\left(\frac{(D+2)(D+4)\sigma^4}{(D-1)\|g(x)\|_2^2}\right).$$

The proof, given in Appendix A.4, shows that $\hat{\sigma}(x)$ tracks the noise level up to higher-order terms. Overall, the uncertainty score $u(x)$ and its derived quantities provide a unified, deterministic description of prediction reliability grounded in the geometry of the output space, without requiring explicit epistemic–aleatoric separation.

## 2.6 EXPERIMENTAL VALIDATION

We empirically validate the proposed uncertainty estimator on a synthetic regression task designed to separately capture *epistemic* and *aleatoric* effects. The ground-truth function is $y = x^3$, with training samples drawn from $x \in [-4, 4]$. Aleatoric uncertainty is induced by adding zero-mean Gaussian noise exclusively within the interval $x \in [-2, 2]$, where the variance increases piecewise-linearly from $\sigma(-2) = 0$ to $\sigma(2) = 20$, as shown by the blue line in Figure 1(b). Outside this interval, the data are noise-free. Epistemic uncertainty is evaluated by probing predictions beyond the training domain, i.e., $x \notin [-4, 4]$.

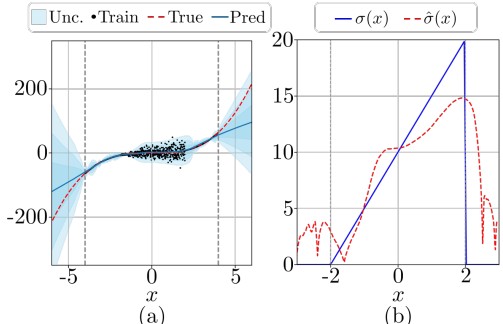

Figure 1: (a) Aleatoric uncertainty is captured in the noisy region ($x \in [-2, 2]$), while epistemic uncertainty appears outside the training domain ($x \notin [-4, 4]$). (b) $\hat{\sigma}(x)$ closely tracks the ground-truth $\sigma(x)$.

**Results.** Figure 1(a) shows regression results with predictive confidence bands ($\pm k\hat{\sigma}(x), k = 1, 2, 3$). Beyond the training domain, the bands widen, reflecting increasing epistemic uncertainty. Within the noisy region $x \in [-2, 2]$, the bands align with the scattered observations, indicating that $\hat{\sigma}(x)$ captures the aleatoric uncertainty. Figure 1(b) further compares the ground-truth $\sigma(x)$ (blue solid) with the estimated $\hat{\sigma}(x)$ (red dashed), confirming that HCM reliably captures the true variance.

## 2.7 THRESHOLDING AND PRACTICAL USE OF THE UNCERTAINTY SCORE.

While the uncertainty score $u(x)$ provides a quantitative indicator of model reliability, using it for decision-making requires a principled way to determine what constitutes "high uncertainty." In practice, an uncertainty score alone is insufficient; users also need a task-dependent criterion that defines when a prediction should be regarded as unreliable. We therefore outline two practical strategies for selecting a threshold on $u(x)$, depending on the structure and requirements of the downstream task.

**Tasks with an explicit accuracy or safety tolerance.** In many real-world applications—particularly industrial or safety-critical settings—there exists a domain-specific tolerance $\varepsilon$ beyond which prediction errors are considered unacceptable. Since the uncertainty score $u(x)$ provides a lower bound on the prediction error, any input satisfying $u(x) > \varepsilon$ can be interpreted as violating this tolerance. In such cases, the threshold is not arbitrary but is directly determined by the operational requirements of the task.

**Tasks without an explicit tolerance.** When no predefined error or safety margin is available, a distributional calibration approach can be applied. A simple and effective guideline is to compute the empirical distribution of $u(x)$ on a validation set and select a high quantile (e.g., 95% or 99%) as the threshold. This identifies the upper tail of uncertainty values and marks predictions whose uncertainty is unusually large compared to typical in-distribution behavior.

## 3 EXPERIMENTS

We conduct a series of experiments to evaluate the effectiveness and generality of our proposed HCM-based uncertainty estimation framework. Our experimental goals are threefold:

- **Conceptual Validation:** Provide intuitive evidence of how the HCM uncertainty score behaves in relation to decision boundaries.

- **Quantitative Benchmarking:** Establish competitive performance against representative baselines on classification and regression benchmarks.

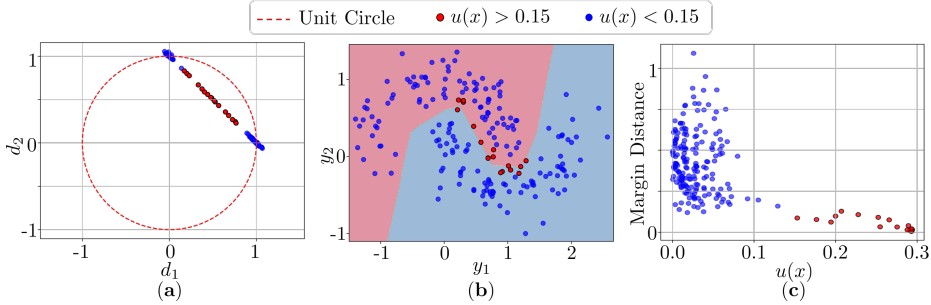

Figure 2: Two-moons experiment. (a) Representation of $\hat{d}$ on the unit circle, with ambiguous samples ($u(x) > 0.15$) shown in red and others in blue. (b) The same samples projected back into the input space. (c) Scatter plot of $u(x)$ against distance to the decision boundary.

- **Practical Applicability:** Assess whether HCM scales to high-dimensional, noisy, and safety-critical industrial data.

We organize the experiments in order of increasing complexity. We begin with classification tasks (Section 3.1) to provide intuition about the behavior of the uncertainty score. We then move to large-scale classification benchmarks (Section 3.2) to test OOD detection performance. Next, we examine calibration for regression benchmarks (Section 3.3) and evaluate high-dimensional industrial regression (Section 3.4) to demonstrate the practical impact. Detailed experimental settings are provided in Appendix A.1.

## 3.1 TWO MOONS

We visually examine how the proposed uncertainty score behaves across the input space, particularly in relation to decision boundaries. For this purpose, we use the two moons binary classification dataset (Pedregosa et al., 2011) and train a HCM classifier with one hidden layer (16 ReLU units). This experiment serves as an intuitive validation of the geometric structure imposed by HCM.

**Results.** Figure 2 illustrates the learned representation from three perspectives. (a) We plot $\hat{d}$ where samples exhibiting high deviation ($u(x) > 0.15$) are highlighted in red. They concentrate along the line connecting the two class directions, $(1, 0)$ and $(0, 1)$, where $\|\hat{d}\|_2$ naturally decreases. (b) Projecting these samples back to the input space reveals that they concentrate near the decision boundary, where the two classes overlap. (c) The scatter plot of $u(x)$ versus distance to the boundary confirms this trend: samples with large $u(x)$ lie close to the decision boundary, while those with small $u(x)$ are located farther away. Taken together, these observations demonstrate that the proposed uncertainty score $u(x)$ serves both as a quantitative measure of prediction confidence and as a qualitative marker of inherently ambiguous regions.

## 3.2 CLASSIFICATION-BASED OOD DETECTION ON CIFAR-10

We now turn to the more challenging setting of multi-class classification to evaluate whether HCM can reliably capture uncertainty on large-scale benchmarks. In particular, we assess HCM for OOD detection under the standardized OpenOOD protocol (Zhang et al., 2023) and compare its performance against a broad range of established OOD detection methods.

We train ResNet-18 (He et al., 2016) on CIFAR-10 (Krizhevsky, 2009) as the in-distribution dataset. To evaluate OOD detection, we consider both near-OOD (semantically related classes) and far-OOD (different modalities) benchmarks. A full list of datasets and references is provided in Appendix A.1.

We compare against a diverse set of representative baselines, including sampling-based approaches (Ensembles (Lakshminarayanan et al., 2017) and MC Dropout (Gal & Ghahramani, 2016)); softmax-based approaches including MSP (Hendrycks & Gimpel, 2016) and ODIN (Liang et al., 2017), which derive OOD scores from softmax logits; and similarity-based approaches such as Energy (Liu et al., 2020), Mahalanobis Distance (MDS) (Lee et al., 2018), and KNN (Sun et al., 2022).

Table 2: AUROC (%) for near- and far-OOD detection across methods. Higher is better. Results are averaged over 5 random seeds.

| OOD Type | MSP | Ensembles | MC | ODIN | Energy | MDS | KNN | ViM | fDBD | NCI | HCM | HCM mix |
|---|---|---|---|---|---|---|---|---|---|---|---|---|
| Near OOD | 86.73 | **88.89** | 85.21 | 85.49 | 87.52 | 84.91 | 88.07 | 84.20 | 88.87 | 86.49 | 82.23 | 87.90 |
| Far OOD | 88.96 | 90.86 | 90.50 | 88.46 | 89.36 | 91.90 | **92.59** | 90.52 | 69.52 | 92.49 | 86.45 | 90.12 |
| AVG | 87.85 | 90.15 | 88.33 | 87.23 | 88.62 | 89.34 | **90.97** | 88.12 | 76.96 | 90.36 | 85.04 | 89.44 |

We also evaluate recent similarity-based variants, including ViM (Wang et al., 2022), fDBD (Liu & Qin, 2023), and NCI (Liu & Qin, 2025), which refine feature-space residuals or density estimation for improved scalability.

**Results.**    Table 2 reports AUROC on the near-OOD and far-OOD datasets. In these experiments, we introduce **HCM mix**, which augments training with interpolated cross-class samples using the mixup technique (He et al., 2019). Vanilla HCM, trained with one-hot supervision, tends to push predictions too strongly toward single class directions, limiting its ability to express uncertainty. Mix-up alleviates this by generating intermediate samples with interpolated labels, which, under the hyperspherical decomposition, correspond to directions lying between class anchors on the unit sphere. These intermediate directions naturally yield its magnitudes below one, enabling HCM to represent uncertainty more faithfully. This hypersphere-based interpretation explains why mixup provides a meaningful boost for HCM but not for conventional methods. In fact, as shown in Appendix A.6, applying mixup to existing baselines does not consistently improve OOD detection and can even degrade performance. In contrast, HCM mix achieves an average AUROC of **89.44%**, competitive with KNN (90.97%) and NCI (90.36%), while attaining the lowest computational latency among all methods. Full per-dataset AUROC, FPR@95TPR, and efficiency results are provided in Appendix A.5.

### 3.3    UNCERTAINTY CALIBRATION FOR DEPTH ESTIMATION

We now turn to regression tasks, where uncertainty calibration remains comparatively underexplored. Unlike classification, which benefits from standardized protocols such as OpenOOD, regression lacks widely adopted benchmarks despite its importance in safety-critical applications (Gustafsson et al., 2023). Because regression uncertainty has received relatively little attention, only a handful of established baselines are available. We therefore compare HCM against three representative approaches: Deep Ensembles (Lakshminarayanan et al., 2017), MC Dropout (Gal & Ghahramani, 2016), and evidential learning (EDL) (Amini et al., 2020).

To ensure comparability across methods, we propose the following calibration procedure on a held-out validation dataset. Raw uncertainty estimates $u$ are scaled using temperature calibration, i.e., $u_{\mathrm{cal}} := uT$, where the scalar $T$ is selected so that the empirical coverage of $\sigma, 2\sigma, 3\sigma$ (where $\sigma$ is the standard deviation of the validation set at each $u_{\mathrm{cal}}$) best approximates the theoretical 68%, 95%, and 99.7% rules. The calibrated uncertainty $u_{\mathrm{cal}}$ is then mapped into confidence values via a negative exponential transformation, $\mathrm{conf}(x) = \exp(-u_{\mathrm{cal}}(x))$. Finally, the confidence values are min–max normalized to $[0, 1]$ to enable fair comparison across methods.

We adopt standard metrics spanning calibration (empirical coverage at $1\sigma$, $2\sigma$, and $3\sigma$; regression Expected Calibration Error, $\mathrm{ECE}_{\mathrm{reg}}$), correlation (Pearson and Spearman), and accuracy (Root Mean Squared Error, RMSE). Together, these metrics provide complementary perspectives on whether confidence scores align with prediction errors at both local and global levels (detailed definitions in Appendix A.2).

We validate HCM in this setting on pixel-wise regression using the NYU-v2 dataset (Silberman et al., 2012) for monocular depth estimation. The backbone is a U-Net–style encoder–decoder (Ronneberger et al., 2015). To systematically increase prediction error variance at test time, each input is perturbed as $x' = x + a\varepsilon$ where $\varepsilon \sim \mathcal{N}(\mathbf{0}, \mathbf{I}) \in \mathbb{R}^{D^*}$ is Gaussian noise and $a \sim \mathcal{U}(0, 0.5)$ is a uniform random variable that scales its amplitude. We also perform regression under distribution shifts on UCI datasets and provide the results in Appendix A.7.

Table 3: Calibration results on NYU-v2 depth estimation.

| Method | cov@$1\sigma$ | cov@$2\sigma$ | cov@$3\sigma$ | Pearson ↑ | Spearman ↑ | ECE$_{reg}$ ↓ | RMSE ↓ |
|---|---|---|---|---|---|---|---|
| EDL | **0.6906** | 0.9252 | 0.9884 | 0.1084 | 0.1370 | **0.0609** | 0.1241 |
| MC Dropout | 0.7019 | 0.9033 | 0.9645 | 0.1932 | 0.2580 | 0.0645 | **0.1189** |
| Ensembles | 0.6957 | **0.9280** | **0.9759** | 0.2381 | 0.4684 | 0.1838 | 0.1234 |
| HCM | 0.7433 | 0.8798 | 0.9167 | **0.4919** | **0.5425** | 0.2160 | 0.1485 |

**Results**  Figure 3 presents the regression results on the test dataset. In (a), HCM exhibits a smooth and consistent reduction in error as confidence increases. This well-behaved trend stems from Proposition 1, which links our uncertainty score directly to the true error. By contrast, the baselines show weaker calibration: EDL provides only limited separation, Ensembles fail to distinguish errors in the low-confidence region, and MC Dropout produces high errors that remain overconfident. In (b), the distribution of confidence scores further highlights these differences: MC Dropout, EDL, and Ensembles collapse predictions into narrow ranges, leading to overconfident behavior, whereas HCM maintains a broad spread across the confidence scale. Finally, (c) illustrates representative examples selected by HCM: high-confidence predictions correspond to clean inputs with small errors, while low-confidence predictions arise from noisy inputs with large errors. Together, these results show that HCM's confidence values meaningfully capture both input quality and prediction reliability.

Table 3 refines these observations. HCM achieves the strongest alignment between predicted uncertainty and realized error—showing the highest Pearson and Spearman correlations. However, HCM lags the variance-based baselines on coverage metrics and ECE$_{reg}$, and also exhibits a higher RMSE. This behavior arises from our decomposition-based prediction: instead of directly regressing the target value, HCM predicts both its magnitude and direction and reconstructs the final output as $Rd$. Small deviations in both components can therefore compound during reconstruction, leading to a modest increase in prediction error compared to direct regression approaches. Nevertheless, the magnitude of this RMSE gap remains limited and does not noticeably degrade overall predictive capability. Despite this slight RMSE gap, HCM provides a more reliable uncertainty score by maintaining a significantly stronger correspondence between uncertainty and actual error.

## 3.4 Industrial High-Dimensional Regression

We consider a real-world industrial regression task to assess the practical applicability of HCM. This experiment uses proprietary data from a semiconductor manufacturing process, where the objective is to predict the 3D geometry of etched wafer holes from spectroscopic ellipsometry measurements and to identify anomalous wafers through reliable uncertainty estimates.

The dataset consists of high-dimensional spectral signals paired with geometric targets, and we train a multi-layer perceptron. Confidence values are obtained by aggregating uncertainties at the sample level, followed by the same exponential transformation and temperature calibration described in Section 3.3. Unlike the regression benchmarks in Section 3.3, however, we apply *quantile normalization* instead of min–max scaling. This produces confidence values that are uniformly distributed

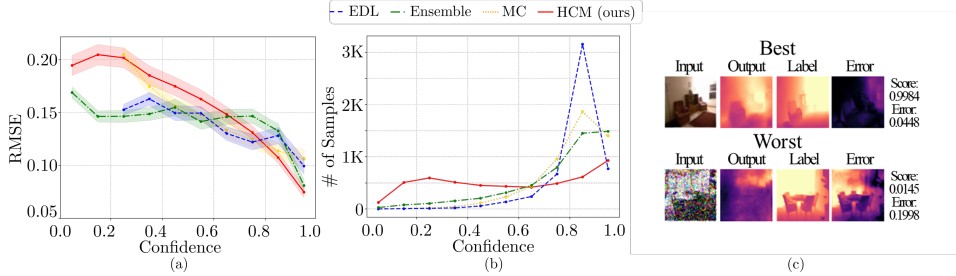

Figure 3: Qualitative and quantitative results for depth estimation. (a) Calibration curves. (b) Distribution of test samples across confidence intervals. (c) Best- and worst-confidence prediction examples with input, prediction, ground truth, and error maps.

Table 4: Calibration results on the industrial regression dataset.

| Method | cov@1$\sigma$ | cov@2$\sigma$ | cov@3$\sigma$ | Pearson ↑ | Spearman ↑ | ECE$_{reg}$ ↓ | RMSE ↓ |
|---|---|---|---|---|---|---|---|
| EDL | 0.6822 | 0.8813 | 0.9453 | $-0.2508$ | $-0.1837$ | **2.2615** | 4.7909 |
| Ensemble | 0.6823 | 0.8755 | 0.9375 | 0.3227 | 0.1220 | 4.1948 | **4.6783** |
| MC Dropout | 0.6789 | **0.8961** | **0.9570** | $-0.0785$ | $-0.0630$ | 4.0164 | 6.5602 |
| HCM | **0.6799** | 0.8667 | 0.9159 | **0.8435** | **0.7579** | 2.3104 | 5.4022 |

across quantile bins, enabling a direct evaluation of whether low-confidence estimates can serve as an effective filter for high-error cases in real-world industrial settings. We also do not inject additional noise at test time, as the industrial signals already contain substantial noise.

**Results.** Figure 4 summarizes the calibration behavior on the industrial dataset. In (a), HCM shows a smooth reduction in error with increasing confidence, reaching the lowest errors in the high-confidence regime while spanning the full confidence range. By contrast, the baselines overall appear uncalibrated, failing to separate high- and low-error cases. In (b), the error distributions appear broadly similar across all methods. Despite this similarity, HCM demonstrates clear discriminative power using its uncertainty score. Finally, (c) compares HCM's error scatter with the theoretical lower bound $u$ from Proposition 1. The figure confirms Proposition 1, which guarantees that when $\epsilon \approx 0$, the uncertainty score $u$ provides a valid lower bound on the prediction error. Empirically, the bound is nearly tight in this dataset, with $u$ closely following errors. This alignment between theory and practice explains why HCM achieves substantially stronger calibration and overall performance than competing methods.

Table 4 presents the quantitative results on the industrial dataset. HCM achieves by far the strongest Pearson and Spearman correlations, demonstrating that its uncertainty scores are most tightly aligned with the true errors. However, HCM attains lower coverage at $2\sigma$ and $3\sigma$, compared to the statistically grounded methods that explicitly estimate variance (EDL, MC Dropout, and Ensembles). RMSE is also higher for HCM, reflecting its sensitivity to the intrinsic noise in the industrial signals, whereas variance-based methods exhibit stronger robustness.

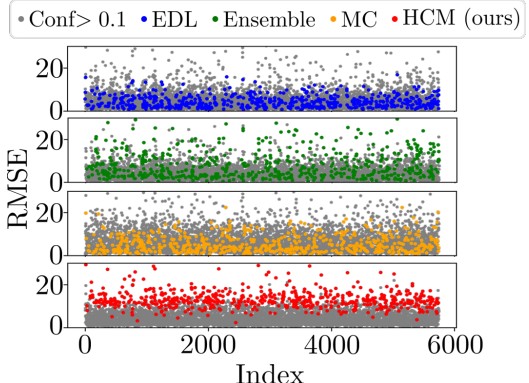

Figure 5: Samples with confidence below 0.1 (colored) and above 0.1 (gray).

Figure 5 highlights the industrial test results across methods. HCM consistently assigns low confidence ($< 0.1$) to samples with large prediction errors, confirming Proposition 1 in practice: when the uncertainty score is large, the error is necessarily large as well. This enables HCM to detect unreliable predictions at test time without access to ground truth, relying solely on its uncertainty score. By contrast, EDL, Ensembles, and

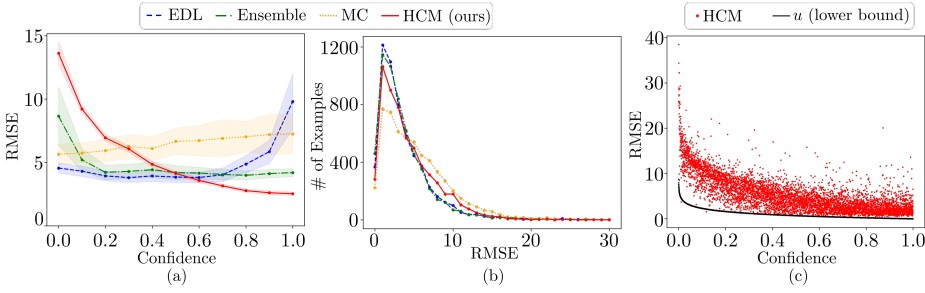

Figure 4: Industrial regression calibration. (a) Calibration curves. (b) Distribution of test samples across prediction error intervals. (c) HCM's RMSE scatter with $u$

MC Dropout struggle to separate high-error cases into low-confidence regions. These findings make HCM particularly well-suited for safety-critical industrial applications, where identifying and filtering out unreliable predictions is crucial for trustworthy deployment.

## 4 CONCLUSION

We introduced HCM, a prior-free and lightweight framework for uncertainty quantification grounded in geometric structure. By aligning uncertainty scores directly with prediction error, HCM provides both theoretical guarantees and strong empirical performance across regression, classification, depth estimation, and industrial datasets. Its core strength lies in geometric decomposition: separating outputs into magnitude and direction yields interpretable scores and enables full use of the confidence spectrum. Unlike variance-based methods prone to overconfidence, HCM maintains fine-grained calibration over the entire [0,1] interval, allowing reliable sample-wise separation.

**Limitations and future work.**

The uncertainty score $u(x)$ may depend on training dynamics. As seen in our ablation study (Appendix A.8), overly large $\lambda$ or an excessively high learning rate can destabilize the unit-norm constraint, pushing $d$ off the hypersphere and hindering the learning of $R$. Thus, $u(x)$ may be influenced by the interaction between constraint strength and optimization settings, and it does not explicitly separate aleatoric from epistemic uncertainty. HCM currently operates only at the output level and requires light fine-tuning, which may limit applicability to frozen or zero-shot large language models. In addition, the method assumes that targets admit a magnitude–direction decomposition, which may restrict tasks with intrinsically multi-valued outputs. Future work includes extending HCM to larger or multimodal models, improving uncertainty decomposition, and leveraging HCM's ability to flag high-error samples for active learning.

## ACKNOWLEDGEMENTS

This research was supported by Samsung Electronic Co., Ltd (IO230425-06044-01).

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

# A  APPENDIX

## A.1  EXPERIMENTAL SETUP

All experiments were conducted on a single NVIDIA GeForce RTX 4090 GPU. The software environment was Python 3.8.10 (GCC 9.4.0) with PyTorch 2.4.0+cu121, running on CUDA 12.1 and cuDNN 9.0.1.

**Two Moons.**  We use the standard two moons dataset with 2-dimensional inputs. The model is a simple MLP consisting of two hidden layers with 16 units each and ReLU activations. It has two output branches: a 2D direction vector $d$ and a scalar magnitude $R$. Training is performed with mean squared error (MSE) loss using the Adam optimizer (learning rate 0.001) for 1000 epochs.

**Classification OOD detection on CIFAR-10.**  We evaluate HCM on large-scale OOD detection using CIFAR-10 as the in-distribution dataset. The backbone is ResNet-18 (He et al., 2016), modified with hyperspherical decomposition into a class-wise direction head and a scalar magnitude head. Training uses SGD with momentum 0.9, weight decay $10^{-4}$, and initial learning rate 0.1 decayed at epochs 30, 60, and 90. We train for 100 epochs with batch size 64.

We adopt MixUp augmentation, supporting both pairwise interpolation ($k = 2$) and Dirichlet-based multi-sample interpolation ($k = 20$, $\alpha = 0.5$). Losses are computed with MSE against soft labels, consistent with the hyperspherical objective.

At test time, we follow the OpenOOD protocol (Zhang et al., 2023) and evaluate on six OOD datasets: **near-OOD datasets** (CIFAR-100 (Krizhevsky, 2009), TinyImageNet (Deng et al., 2009)) and **far-OOD dataset** (MNIST (LeCun et al., 1998), SVHN (Netzer et al., 2011), Texture (Cimpoi et al., 2014), Places365 (Zhou et al., 2018)). We report AUROC and FPR@95TPR.

In addition, we instrument runtime efficiency by measuring parameter count, FLOPs/MACs, pure inference latency/throughput (batch sizes 1 and 64), GPU peak memory usage, and HCM scoring cost (average milliseconds per image).

**Uncertainty Calibration for Depth Estimation.**  We further validate HCM on pixel-wise regression using the NYU-v2 dataset (Silberman et al., 2012) for monocular depth estimation. We construct paired image–depth datasets from official training splits, and resize all images to $64 \times 64$. The data is divided into 80% training, 10% validation, and 10% testing.

The backbone is a U-Net style encoder–decoder (Ronneberger et al., 2015), denoted HCM-UNet, with four downsampling blocks, a bottleneck, and four upsampling blocks. At the output stage, the network predicts both a pixel-wise direction map $d$ and a magnitude map $R$, whose product reconstructs the depth image.

Training is performed with mean squared error (MSE) loss for 200 epochs, using the Adam optimizer (learning rate $2 \times 10^{-4}$, betas $(0.9, 0.999)$) and a multi-step scheduler that decays the learning rate by 0.1 at epochs 50, 100, and 150. All weights are initialized from a normal distribution.

At test time, Gaussian noise $a \cdot \mathcal{N}(0, 1)$ with $a \in [0, 0.5]$ is added to input images to probe robustness. Evaluation metrics include sample-wise RMSE, uncertainty–error scatter plots, selective prediction metrics, and correlation (Spearman's $\rho$) between confidence scores and errors. In addition, we visualize best- and worst-confidence predictions, and report error distributions across confidence bins using box plots.

**Industrial High-Dimensional Regression.** We further evaluate HCM on a real-world semiconductor manufacturing dataset, where the task is to predict the 3D geometry of etched wafer holes from high-dimensional spectral inputs. The dataset consists of 1,375 training samples and 4,596 test samples. Each input is a 400-dimensional spectrum signal, and the target is a 25-dimensional geometry vector.

The regression model is a fully connected network with three hidden layers of width 200, 100, and 50, each followed by a LeakyReLU activation (negative slope 0.01). The network produces two outputs: a scalar magnitude $R$ and a 25-dimensional direction vector $d$. Training is performed with mean squared error (MSE) loss for 1000 epochs, using the Adam optimizer with learning rate 0.002, betas $(0.9, 0.999)$, and weight decay $10^{-4}$. The learning rate is scheduled to decay by a factor of 0.1 at epochs 150, 300, 450, 600, 750, and 900.

**UCI benchmark regression.** We evaluate HCM on six standard UCI regression datasets obtained from OpenML (Vanschoren et al., 2013): *Wine Quality*, *Concrete Strength*, *Energy Efficiency*, *Kin8nm*, *Power Plant*, and *Yacht Hydrodynamics*. Each dataset is split into 80% training 10% validation and 10% testing with a fixed random seed for reproducibility.

The model is a fully connected network with three hidden layers of 20 units each and LeakyReLU activations (negative slope 0.01). It outputs both a scalar magnitude $R$ and a low-dimensional direction vector $d$. Training is performed with mean squared error (MSE) loss for 200 epochs using the Adam optimizer (learning rate $10^{-4}$, betas $(0.9, 0.999)$, weight decay $10^{-4}$).

At test time, we inject additive Gaussian noise $\epsilon \sim \mathcal{N}(0, \sigma^2 I)$ into the input features, with $\sigma \in 0, 1, 2, 3, 4, 5$. This protocol evaluates both the global sensitivity of uncertainty scores under increasing perturbations and their per-sample calibration at fixed noise levels.

**Text Classification OOD Detection.** We evaluate the extensibility of HCM to natural-language classification using AG News (Zhang et al., 2015) as the in-distribution dataset. A bert-base-uncased encoder (Devlin et al., 2019) is first fine-tuned on the AG News training set, after which the classifier head is replaced with the HCM decomposition: a direction head $d \in \mathbb{R}^4$ and a scalar magnitude head $R \in \mathbb{R}$ applied to the CLS representation. The model is then trained for one epoch using the HCM loss with batch size 16, maximum sequence length 128, and learning rate $2 \times 10^{-6}$.

To assess OOD detection, we adopt a standard binary ID–OOD evaluation protocol using four text datasets: *Yelp Polarity*, *IMDB*, *Emotion*, and *DBPedia-14* (Zhang et al., 2015; Maas et al., 2011; Saravia et al., 2018). For each dataset, we compute AUROC, FPR@95TPR, and ID accuracy on AG News. All results are averaged over five independent random seeds.

**Additional 1D Regression.** To further examine how hyperspherical decomposition behaves in the scalar regression setting, we conduct a series of controlled 1D experiments using synthetic datasets with diverse noise structures: heteroscedastic Gaussian noise, heteroscedastic Laplace noise, bimodal Gaussian mixture noise, and multi-valued regression ($y = \pm\sqrt{x}$). For each setting, we train a lightweight fully connected network that predicts a 2D direction vector $d$ and a scalar magnitude $R$, where scalar targets are embedded into a 2D space via duplication $(y, y)$ to enable the same decomposition as in higher-dimensional regression. All models are trained for 1000 epochs

using Adam (learning rate $10^{-3}$) with a multi-step schedule, and we report both prediction quality and uncertainty behavior across noise regimes. To contextualize HCM's behavior, we compare against representative uncertainty baselines: **Mixture Density Networks (MDN)**(El-Laham et al., 2023), **Bayesian Neural Networks (BNN)**(Sun et al., 2017) with learned epistemic uncertainty, and **Bayesian** models with a global aleatoric noise parameter. All models use comparable network capacity (three hidden layers of width 100) and are trained for 1000 epochs using Adam with learning rate $10^{-3}$ and a multi-step decay schedule. For each noise regime, we report predictive fit and per-sample uncertainty behavior.

## A.2 REGRESSION CALIBRATION METRICS

We briefly describe the metrics used in our regression experiments:

Let $\mathcal{D} = \{(x_i, y_i)\}_{i=1}^{N}$ be the evaluation set. The model produces predictions $\hat{y}_i$ and an associated uncertainty $u_i$. After temperature calibration, we obtain

$$u_{\text{cal},i} = T \cdot u_i,$$

and define the per-sample error $r_i$ as the RMSE of prediction $\hat{y}_i$ with respect to the target $y_i$. Confidence values are further computed as $c_i = \exp(-u_{\text{cal},i})$, but the metrics are based on $(u_{\text{cal}}, r)$.

**Coverage at $1\sigma, 2\sigma, 3\sigma$.** We measure the empirical coverage at three scales:

$$\text{Cov}@k\sigma = \frac{1}{N} \sum_{i=1}^{N} \mathbf{1}\{\, r_i \leq k\, u_{\text{cal},i} \,\}, \quad k \in \{1, 2, 3\}.$$

The desired values are $(0.68, 0.95, 0.997)$ according to the Gaussian 68–95–99.7 rule. Closer is better.

**Regression Expected Calibration Error (ECE$_{\text{reg}}$).** Samples are partitioned into $B$ bins according to $u_{\text{cal}}$. For bin $b$, let $S_b = \{i \mid \tau_{b-1} \leq u_{\text{cal},i} < \tau_b\}$, and define

$$\bar{u}_b = \frac{1}{|S_b|} \sum_{i \in S_b} u_{\text{cal},i}, \qquad \bar{r}_b = \frac{1}{|S_b|} \sum_{i \in S_b} r_i.$$

Then

$$\text{ECE}_{\text{reg}} = \frac{1}{\sum_b |S_b|} \sum_{b=1}^{B} |S_b| \cdot \left| \bar{u}_b - \bar{r}_b \right|.$$

A smaller value indicates better calibration.

**Correlation.** We report both Pearson correlation,

$$\rho_{\text{Pearson}} = \frac{\sum_{i=1}^{N} (u_{\text{cal},i} - \bar{u}_{\text{cal}})(r_i - \bar{r})}{\sqrt{\sum_{i=1}^{N} (u_{\text{cal},i} - \bar{u}_{\text{cal}})^2} \sqrt{\sum_{i=1}^{N} (r_i - \bar{r})^2}},$$

and Spearman correlation,

$$\rho_{\text{Spearman}} = \rho_{\text{Pearson}}\big(\text{rank}(u_{\text{cal}}),\ \text{rank}(r)\big).$$

Higher is better, reflecting stronger alignment between uncertainty and error.

**Root Mean Squared Error (RMSE).** Finally, we report accuracy as the average per-sample RMSE:

$$\text{RMSE} = \frac{1}{N} \sum_{i=1}^{N} r_i.$$

Lower is better.

## A.3 DETAILS OF THE TRAINING OBJECTIVE

From the constrained formulation in (1), expanding the prediction error with $e_y = \hat{R}e_d + e_R d$ yields

$$\|e_y\|_2^2 = \|\hat{R}e_d\|_2^2 + e_R^2 + 2\hat{R}e_R\langle e_d, d\rangle,$$

which contains a directional term weighted by $\hat{R}^2$ as well as the cross term $2\hat{R}e_R\langle e_d, d\rangle$. In our training loss in Section 2.4), we instead use the ground-truth magnitude $R^2$ to weight the directional error:

$$\mathcal{L}_{\text{dir}} = R^2\|e_d\|_2^2.$$

Since $\hat{R}^2 = R^2 + 2Re_R + e_R^2$, near the optimum ($\hat{R} \approx R$) the difference between using $\hat{R}^2$ and $R^2$ is only $O(|e_R|)$.

## A.4 PROOF DETAILS FOR PROPOSITION 2

As defined in Section 2.4, the total training objective is

$$\mathcal{L}_{\text{total}} = \phi_d\big(R\,\|e_d\|_2\big) \;+\; \phi_R\big(e_R\big) \;+\; \lambda_{\text{norm}}\,\phi_{\text{norm}}\big(\,\big|\|\hat{d}\|_2 - 1\big|\,\big).$$

Excluding the norm-regularization term, the loss naturally decomposes into a direction part $\phi_d$ and a magnitude part $\phi_R$. For the proof, it is convenient to analyze the two branches separately.

**Step 1: Magnitude branch.** Let $y = g(x) + \varepsilon$, where the ground truth $y$ is decomposed into the deterministic part $g(x)$ and the noise $\varepsilon \sim \mathcal{N}(0, \sigma^2 I_D)$. Since $\mathbb{E}\big[\|\varepsilon\|_2^2\big] = D\sigma^2$ and $\mathbb{E}\langle g(x), \varepsilon\rangle = 0$, we obtain

$$\mathbb{E}\big[R^2\big] = \mathbb{E}\big[\|y\|_2^2\big] = \mathbb{E}\big[\|g(x)\|_2^2 + 2\langle g(x), \varepsilon\rangle + \|\varepsilon\|_2^2\big] = \|g(x)\|_2^2 + D\sigma^2.$$

Hence, the minimizer of the magnitude branch is

$$\hat{R}^\star(x)^2 = \|g(x)\|_2^2 + D\sigma^2.$$

**Step 2: Direction branch.** For the direction branch, we focus on the normalized target

$$d(x) = \frac{y}{\|y\|_2} = \frac{g(x) + \varepsilon}{\|g(x) + \varepsilon\|_2}, \qquad \varepsilon \sim \mathcal{N}(0, \sigma^2 I_D).$$

As discussed above, it is natural to interpret $d(x)$ through the von Mises–Fisher (vMF) distribution (Mardia & Jupp, 2009): we view it as a random unit vector whose mean direction is

$$\mu(x) \;=\; \frac{g(x)}{\|g(x)\|_2}.$$

For a vMF-distributed random unit vector $d$, the expectation always has the form

$$\mathbb{E}[d] \;=\; A_D(\kappa)\,\mu,$$

i.e., it is a scalar multiple of the mean direction. In our setting, this implies that the expectation of the normalized noisy target must be of the form

$$\hat{d}^*(x) = \mathbb{E}\left[\frac{y}{\|y\|_2}\right] \;=\; C\,\frac{g(x)}{\|g(x)\|_2},$$

for some alignment coefficient $C \in (0, 1]$ that depends on the noise level and the dimension $D$.

To compute $C$, we project $d(x)$ onto the true direction $\mu(x)$ and take expectation:

$$C = \mu(x)^\top \mathbb{E}[d(x)] = \mathbb{E}\big[\mu(x)^\top d(x)\big] = \mathbb{E}\left[\frac{y}{\|y\|_2} \cdot \frac{g(x)}{\|g(x)\|_2}\right] = \frac{1}{\|g(x)\|_2}\mathbb{E}\left[\frac{y \cdot g(x)}{\|y\|_2}\right].$$

Thus the problem of characterizing the vector expectation reduces to computing the scalar quantity $\mathbb{E}[y \cdot g(x)/\|y\|_2]$.

Using the expansion

$$\frac{1}{\|g(x) + \varepsilon\|_2} = \frac{1}{\|g(x)\|_2}\left(1 + \frac{2\,g(x)\cdot\varepsilon}{\|g(x)\|_2^2} + \frac{\|\varepsilon\|_2^2}{\|g(x)\|_2^2}\right)^{-1/2},$$

and a Taylor series up to order $\sigma^4$, we obtain

$$\mathbb{E}\left[\frac{y \cdot g(x)}{\|y\|_2}\right] = \|g(x)\|_2 + \frac{(1 - D)\sigma^2}{2\|g(x)\|_2} + O\left(\frac{(D + 2)(D + 4)\sigma^4}{\|g(x)\|_2^3}\right).$$

Dividing by $\|g(x)\|_2$ gives

$$C = 1 + \frac{(1 - D)\sigma^2}{2\|g(x)\|_2^2} + O\left(\frac{(D + 2)(D + 4)\sigma^4}{\|g(x)\|_2^4}\right).$$

Our uncertainty measure for the direction branch is based on the deviation of the squared norm of the mean direction from 1, namely

$$1 - \|\hat{d}^*(x)\|_2^2 = 1 - \left\|\mathbb{E}\left[\frac{y}{\|y\|_2}\right]\right\|_2^2 = 1 - C^2.$$

Using the above expansion of $C$, we obtain

$$1 - C^2 = \frac{(D - 1)\sigma^2}{\|g(x)\|_2^2} + O\left(\frac{(5D^2 + 22D + 33)\sigma^4}{4\|g(x)\|_2^4}\right).$$

—

**Step 3: Product form.**   Combining the results from Step 1 and Step 2, we multiply the magnitude and direction estimates. From Step 1,

$$\hat{R}^2 = \|g(x)\|_2^2 + D\sigma^2,$$

and from Step 2,

$$1 - \|\hat{d}\|_2^2 = \frac{(D - 1)\sigma^2}{\|g(x)\|_2^2} + O\left(\frac{(D + 2)(D + 4)\sigma^4}{\|g(x)\|_2^4}\right).$$

Thus,

$$\hat{R}^2\left|1 - \|\hat{d}\|_2^2\right| = (D - 1)\sigma^2 + O\left(\frac{(9D^2 + 18D)\sigma^4}{4\|g(x)\|_2^2} + \frac{D(5D^2 + 22D + 33)\sigma^6}{4\|g(x)\|_2^4}\right).$$

Dividing by $D - 1$ gives

$$\frac{\hat{R}^2\left|1 - \|\hat{d}\|_2^2\right|}{D - 1} = \sigma^2 + O\left(\frac{(D + 2)(D + 4)\sigma^4}{(D - 1)\|g(x)\|_2^2} + \frac{D(D + 2)(D + 4)\sigma^6}{(D - 1)\|g(x)\|_2^4}\right).$$

Keeping only the dominant remainder term yields the simplified form

$$\frac{\hat{R}^2\left|1 - \|\hat{d}\|_2^2\right|}{D - 1} = \sigma^2 + O\left(\frac{(D + 2)(D + 4)\sigma^4}{(D - 1)\|g(x)\|_2^2}\right).$$

Therefore, our estimator satisfies

$$\hat{\sigma}^2(x) = \frac{\hat{R}^2\left|1 - \|\hat{d}\|_2^2\right|}{D - 1} = \tfrac{1}{D-1}\,u(x)\left(\hat{R}(x)(1 + \|\hat{d}(x)\|_2)\right) = \sigma^2 + O\left(\frac{(D + 2)(D + 4)\sigma^4}{(D - 1)\|g(x)\|_2^2}\right).$$

Table 5: AUROC (%) for near- and far-OOD detection across different datasets. Higher is better.

| Method | CIFAR-100 | TIN | MNIST | SVHN | Texture | Places365 | AVG |
|---|---|---|---|---|---|---|---|
| MSP | 86.97 | 86.49 | 90.67 | 88.83 | 87.89 | 87.46 | 88.05 |
| Ensemble | 89.16 | 88.62 | 92.06 | 90.52 | 91.63 | 89.23 | 90.20 |
| MC Dropout | 85.30 | 85.12 | 93.81 | 91.07 | 85.76 | 89.34 | 88.40 |
| ODIN | 85.51 | 85.47 | 99.18 | 82.37 | 83.07 | 88.21 | 87.30 |
| Energy | 87.34 | 87.69 | 97.41 | 85.38 | 85.52 | 89.12 | 88.74 |
| MDS | 84.92 | 84.89 | 92.03 | 94.31 | 94.36 | 86.90 | 89.55 |
| KNN | 88.29 | 87.84 | 95.10 | 93.48 | 93.29 | 88.47 | 91.26 |
| ViM | 84.18 | 84.22 | 92.29 | 90.10 | 93.75 | 85.94 | 88.41 |
| fDBD | 88.86 | 88.87 | 94.44 | 65.52 | 57.27 | 62.86 | 76.30 |
| NCI | 86.57 | 86.40 | 95.96 | 92.21 | 93.14 | 88.64 | 90.49 |
| HCM | 81.89 | 82.57 | 85.64 | 89.95 | 87.09 | 83.12 | 85.04 |
| HCM mix | 87.80 | 87.99 | 89.81 | 92.61 | 90.83 | 88.02 | 89.51 |

Table 6: FPR@95TPR (%) for near- and far-OOD detection across different datasets. Lower is better.

| Method | CIFAR-100 | TIN | MNIST | SVHN | Texture | Places365 | AVG |
|---|---|---|---|---|---|---|---|
| MSP | 53.08 | 43.27 | 23.64 | 25.82 | 34.96 | 42.47 | 37.21 |
| Ensemble | 33.73 | 38.23 | 23.73 | 26.51 | 24.86 | 34.60 | 30.28 |
| MC Dropout | 43.80 | 47.02 | 26.91 | 28.62 | 38.45 | 33.88 | 36.45 |
| ODIN | 54.22 | 58.00 | 4.04 | 53.15 | 51.63 | 47.74 | 46.28 |
| Energy | 46.63 | 49.25 | 10.33 | 43.19 | 50.89 | 44.13 | 40.74 |
| MDS | 56.38 | 57.31 | 27.42 | 22.84 | 29.61 | 52.50 | 41.01 |
| KNN | 40.46 | 44.87 | 15.36 | 19.19 | 25.59 | 41.41 | 31.15 |
| ViM | 58.96 | 59.63 | 25.88 | 36.27 | 32.59 | 55.67 | 44.83 |
| fDBD | 53.20 | 50.17 | 35.00 | 92.81 | 84.78 | 92.44 | 68.07 |
| NCI | 57.11 | 60.67 | 16.59 | 24.00 | 26.15 | 47.57 | 38.68 |
| HCM | 79.24 | 77.81 | 69.56 | 46.78 | 66.12 | 76.94 | 69.40 |
| HCM mix | 48.30 | 50.83 | 42.19 | 21.06 | 37.05 | 53.27 | 42.11 |

## A.5 DETAILED OOD DETECTION RESULTS

This section provides detailed results complementing the CIFAR-10 experiments in Section 3.2. While the main text reports only average AUROC for near- and far-OOD benchmarks, here we present per-dataset AUROC and FPR@95TPR (Tables 5 and 6). We further include an efficiency benchmark (Table 7), which compares the computational overhead, latency, throughput, and memory usage of different methods. These results demonstrate that HCM achieves competitive efficiency while maintaining reliable OOD detection performance.

Table 5 reports AUROC values for each individual OOD dataset. While the main text focused on average performance across near- and far-OOD groups, these per-dataset results provide a finer-grained view. HCM by itself underperforms compared to several specialized baselines, but when combined with the proposed mixing strategy (HCM mix), performance improves substantially and becomes competitive across diverse OOD types. For example, AUROC gains are particularly clear on CIFAR-100 and SVHN, indicating that the mixing strategy helps in both semantically similar (near-OOD) and dissimilar (far-OOD) settings.

Table 6 presents FPR@95TPR, a stricter metric that highlights false positive behavior under high recall. The results confirm the trend seen in AUROC: HCM alone struggles on some datasets, especially near-OOD cases, but HCM mix significantly reduces false positives. Notably, the improvements are consistent across both image-based (e.g., TinyImageNet, Places365) and texture-style OOD data, demonstrating the robustness of the approach.

Table 7: Efficiency benchmark of OOD detection methods. All results measured on CIFAR-10 backbone (ResNet). Latency and throughput are measured on ID subset.

| Method | Extra Cost | Latency (ms/img) | Throughput (img/s) | Peak Mem (MB) |
|---|---|---|---|---|
| MSP | Softmax only | 0.15–0.20 | 6k–7k | 227 |
| Ensemble (M=5) | $5\times$ forward passes | 0.25–0.38 | 2.6k–4.3k | 577 |
| MC Dropout | $50\times$ stochastic passes | 1.6–1.8 | $\sim$600 | 227 |
| ODIN | MSP + backward wrt input | 0.36–0.66 | 1.5k–2.8k | 452 |
| Energy | log-sum-exp energy | 0.16–0.33 | 3k–6k | 227 |
| MDS | feature + Mahalanobis score | 0.20–0.36 | 4.9k–5.9k | 228 |
| KNN | feature + KNN search | 0.51–0.66 | 1.5k–2.0k | 227 |
| VIM | feature + covariance eigenspace | 0.18–0.20 | 5.4k–5.5k | 227 |
| fDBD | feature + distance-based score | 0.21 | 4.7k–6.3k | 228 |
| NCI | feature + canonical corr score | 0.37 | 2.7k–6.5k | 228 |
| HCM (ours) | magnitude + direction calc | 0.19 | 5k–6.8k | 226 |

Finally, Table 7 evaluates efficiency trade-offs. Here, sampling-based approaches such as Ensembles and MC Dropout incur substantial costs in latency, throughput, and memory. In contrast, HCM achieves a favorable balance: its runtime and memory footprint are nearly identical to MSP or Energy, yet it offers stronger calibration and competitive OOD detection performance. This efficiency advantage makes HCM particularly suitable for deployment in real-time or resource-constrained environments.

## A.6 EFFECT OF MIX-UP ON OOD DETECTION

Table 8 and Table 9 report the OOD detection performance of various methods when trained with mix-up. We observe that, in general, applying mix-up to existing baselines does not consistently improve OOD detection performance, and in some cases even degrades it. For example, MSP, ODIN, and Energy all suffer substantial drops in AUROC ($-3.61\%$, $-20.95\%$, and $-15.34\%$, respectively), while their FPR@95TPR values increase sharply. Other baselines such as Ensemble, MDS, and KNN show moderate gains (e.g., KNN improves AUROC by $+2.54\%$ and reduces FPR by $-3.31\%$), but these changes remain relatively small.

In contrast, our proposed HCM benefits significantly from mix-up: thanks to its hyperspherical decomposition, mix-up enforces linear relations between interpolated samples on the unit sphere, leading to a meaningful gain in OOD detection performance. Specifically, HCM achieves an average AUROC improvement of $+6.38\%$ and a reduction in FPR@95TPR by $-37.67\%$, representing the largest improvement among all methods. Notably, HCM's AUROC consistently increases across all OOD datasets, while its FPR decreases markedly, especially on SVHN and MNIST.

These results highlight that the improvement under mix-up is not universal, but rather a structural advantage of HCM. The hyperspherical formulation uniquely aligns with the interpolation principle of mix-up, allowing OOD detection performance to scale with the regularization. By contrast, other methods either lack this geometric structure or rely on variance-based uncertainty estimates that are not directly compatible with mix-up. Overall, the findings demonstrate that mix-up is most effective when combined with HCM's representation, and does not yield general benefits across existing methods.

## A.7 UCI DATASET RESULTS

To further validate the generality of our approach, we also report results on six UCI regression benchmarks: Concrete Strength, Energy Efficiency, Kin8nm, Power Plant, Wine Quality, and Yacht Hydrodynamics. In these experiments, Gaussian noise $\mathcal{N}(0, 5I)$ is injected into the test inputs to induce distribution shifts, allowing us to systematically examine the robustness of uncertainty estimates. Specifically, we evaluate whether the proposed method can reliably detect such shifts and whether calibration performance is preserved under distributional perturbations. These additional experiments complement the main results by confirming that our method maintains reliable calibration and uncertainty estimation across diverse datasets and under noisy input conditions.

Figure 6 presents the behavior of each method under distribution shifts on the six UCI regression datasets: Concrete Strength, Energy Efficiency, Kin8nm, Power Plant, Wine Quality, and Yacht Hy-

Table 8: AUROC (%) results on CIFAR-10 (ID) with multiple OOD datasets. Higher is better. **Diff** denotes the change from vanilla to mix-up.

| Method | CIFAR-100 | SVHN | DTD | TIN | MNIST | Places365 | AVG | Diff |
|---|---|---|---|---|---|---|---|---|
| MSP | 83.93 | 83.84 | 82.97 | 85.09 | 85.25 | 85.57 | 84.44 | −3.61 |
| Ensemble | 89.96 | 92.28 | 92.12 | 89.36 | 93.33 | 89.90 | 91.16 | +0.96 |
| MC Dropout | 85.12 | 84.62 | 83.50 | 84.44 | 95.57 | 86.44 | 86.61 | −1.79 |
| ODIN | 66.28 | 39.11 | 74.58 | 69.35 | 69.80 | 79.00 | 66.35 | −20.95 |
| Energy | 75.48 | 55.39 | 74.77 | 79.91 | 75.56 | 79.29 | 73.40 | −15.34 |
| MDS | 85.57 | 96.42 | 90.01 | 84.86 | 99.19 | 91.86 | 91.31 | +1.76 |
| KNN | 89.73 | 96.02 | 94.42 | 89.67 | 96.99 | 90.02 | 93.80 | +2.54 |
| VIM | 82.00 | 92.61 | 95.23 | 83.49 | 92.15 | 86.54 | 88.67 | +0.26 |
| fDBD | 25.82 | 25.03 | 21.97 | 25.06 | 5.72 | 19.34 | 20.49 | −55.81 |
| NCI | 74.02 | 86.19 | 82.69 | 75.23 | 96.65 | 77.49 | 82.04 | −8.45 |
| HCM | 88.31 | 92.31 | 90.93 | 88.44 | 93.06 | 88.76 | 90.30 | +5.26 |

Table 9: FPR@95TPR (%) on CIFAR-10 (ID) with multiple OOD datasets. Lower is better. **Diff** denotes the change from vanilla to mix-up.

| Method | CIFAR-100 | SVHN | DTD | TIN | MNIST | Places365 | AVG | Diff |
|---|---|---|---|---|---|---|---|---|
| MSP | 75.37 | 79.44 | 87.76 | 69.62 | 87.08 | 71.47 | 78.12 | +40.91 |
| Ensemble | 31.99 | 22.71 | 24.25 | 37.84 | 19.57 | 33.85 | 28.37 | −1.91 |
| MC Dropout | 28.70 | 21.55 | 25.90 | 27.43 | 20.71 | 23.69 | 24.33 | −12.12 |
| ODIN | 86.24 | 95.37 | 91.80 | 83.56 | 96.92 | 79.57 | 88.24 | +41.96 |
| Energy | 85.69 | 95.54 | 92.14 | 80.40 | 93.06 | 84.07 | 88.15 | +47.41 |
| MDS | 60.77 | 13.95 | 39.28 | 63.03 | 3.11 | 49.51 | 38.44 | −2.57 |
| KNN | 38.01 | 12.65 | 21.20 | 42.71 | 10.05 | 42.44 | 27.84 | −3.31 |
| VIM | 70.04 | 29.28 | 25.15 | 65.51 | 33.15 | 57.31 | 46.74 | +1.91 |
| fDBD | 99.82 | 99.66 | 99.88 | 99.84 | 99.99 | 99.90 | 99.85 | +31.78 |
| NCI | 87.78 | 59.66 | 75.01 | 86.61 | 16.97 | 81.91 | 67.99 | +29.31 |
| HCM | 44.14 | 19.87 | 32.83 | 46.49 | 27.38 | 46.11 | 36.47 | −27.29 |

drodynamics. All methods were first temperature-scaled on the validation set so that the minimum confidence was at least 0.9. For evaluation, Gaussian noise $\mathcal{N}(0, 5I)$ was injected into the test set inputs to induce distribution shifts, with the black dashed line in the figure denoting the boundary between validation and test data.

Across all six datasets, HCM consistently exhibited a clear drop in confidence once the distribution shift occurred. This demonstrates that HCM not only provides well-calibrated uncertainty estimates on clean data but also responds sensitively to shifts, thereby capturing distributional changes effectively. By contrast, Ensemble showed partial sensitivity, with confidence decreases observed only in datasets such as Concrete Strength, Energy Efficiency, and Kin8nm, while remaining largely unchanged in the others. MC Dropout and EDL performed the worst: despite the added noise, their confidence values exhibited little to no variation, indicating poor capability to detect shifts.

These results confirm that HCM goes beyond achieving reliable calibration under standard conditions. It also generalizes across diverse regression benchmarks by effectively detecting distribution shifts, highlighting its utility in safety-critical applications where robustness to input perturbations is essential.

Table 10 reports detailed results on six UCI regression datasets (Concrete Strength, Energy Efficiency, Kin8nm, Power Plant, Wine Quality, and Yacht Hydrodynamics) under domain shifts induced by Gaussian noise $\mathcal{N}(0, 5I)$. The evaluation covers calibration quality (Monotonicity Index, Isotonic Calibration Error, correlation coefficients, and E-AURC) as well as predictive accuracy (RMSE).

Across most datasets, HCM consistently delivers the strongest calibration performance. On *Energy* and *Kin8nm*, HCM clearly dominates, achieving the highest MI and correlation scores while yielding

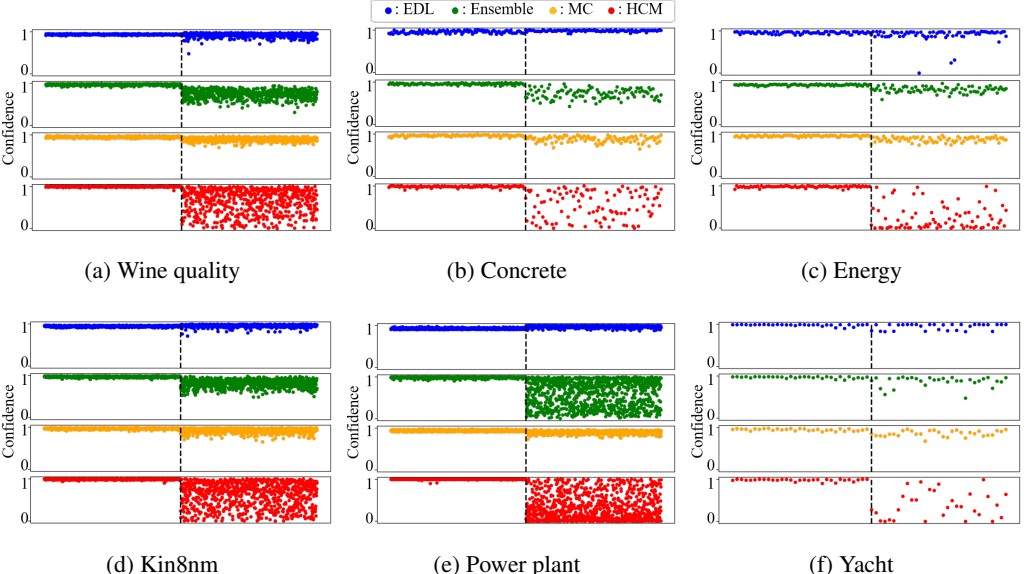

Figure 6: Distribution shift detection on six UCI regression datasets (Concrete Strength, Energy Efficiency, Kin8nm, Power Plant, Wine Quality, Yacht Hydrodynamics). The black dashed line separates validation (left) and test (right) data. At test time, Gaussian noise $\mathcal{N}(0, 5I)$ was injected to induce distribution shifts. Confidence values were temperature-scaled on the validation set (minimum confidence $\geq 0.9$). HCM consistently shows a clear drop in confidence under noisy test inputs, effectively capturing distribution shifts, whereas Ensemble exhibits only partial sensitivity and MC Dropout/EDL remain largely insensitive.

the lowest ICE and E-AURC. On *Power Plant*, HCM achieves perfect monotonicity (MI = 1.0) with exceptionally high correlation coefficients, demonstrating its ability to capture uncertainty under severe distributional shifts. Although HCM sometimes shows relatively larger RMSE values (e.g., *Concrete*, *Yacht*), this behavior is expected given that HCM explicitly prioritizes calibration and shift sensitivity over raw error minimization.

By comparison, MC Dropout often performs well in terms of RMSE and occasionally ICE (for instance, on *Concrete* and *Wine Quality*), but its calibration metrics are inconsistent across datasets. Ensemble methods sometimes reduce RMSE (notably on *Concrete* and *Wine Quality*) but generally fail to provide robust calibration, while EDL struggles both in correlation and in calibration quality.

Taken together, these results confirm that HCM generalizes reliably across diverse regression tasks. It consistently provides well-calibrated uncertainty estimates, responds sensitively to distribution shifts, and maintains competitive predictive accuracy, making it particularly suitable for safety-critical applications where robustness under input perturbations is essential.

## A.8 Ablation on $\lambda$ on UCI dataset

To examine the sensitivity of the Lagrange multiplier $\lambda$, we conduct an ablation study on six UCI regression datasets. In this experiment, we vary $\lambda \in \{0, 1, 3, 5\}$ and report the validation MAE for each configuration. This setup allows us to isolate the effect of $\lambda$ on predictive performance and to identify whether the model exhibits meaningful sensitivity to the choice of this parameter.

Table 11 shows that across the six UCI regression datasets, the model exhibits only weak sensitivity to the choice of $\lambda$ in several cases. In particular, the Wine, Yacht, and Concrete datasets display only minor fluctuations in validation MAE as $\lambda$ varies. This pattern suggests that the unit-norm constraint is already well satisfied during training for these tasks, resulting in a small contribution from the constraint term and thus reducing the overall influence of $\lambda$.

For the Energy and Power Plant datasets, the variation across different $\lambda$ values is larger, but no consistent monotonic trend is present. Finally, the Kin8nm dataset shows a notable degradation when

Table 10: Regression benchmarks under domain shift. We inject Gaussian noise $\mathcal{N}(0, 5I)$ into the input features to simulate distributional shift. We report Monotonicity Index (MI, higher is better), Isotonic Calibration Error (ICE, lower is better), correlation coefficients (Pearson, Spearman, higher is better), E-AURC (lower is better), and overall mean RMSE (lower is better). Best values are in **bold**, second best are underlined.

| Dataset | Method | MI | ICE | Pearson | Spearman | E-AURC | RMSE |
|---------|--------|-----|-----|---------|----------|--------|------|
| Concrete | EDL | 0.636 | 38.93 | 0.196 | 0.256 | 15.86 | **43.50** |
| | MC Dropout | **0.867** | **29.14** | **0.761** | **0.785** | **7.46** | 75.03 |
| | Ensemble | 0.533 | 41.17 | 0.214 | 0.229 | 25.38 | 57.79 |
| | HCM | 0.611 | 37.42 | 0.760 | 0.667 | 13.47 | 76.33 |
| Energy | EDL | 0.429 | 30.01 | 0.498 | 0.630 | 9.95 | 45.82 |
| | MC Dropout | 0.786 | 17.23 | 0.752 | 0.773 | 5.12 | 45.89 |
| | Ensemble | 0.300 | 31.29 | -0.056 | -0.094 | 22.64 | 33.35 |
| | HCM | **0.842** | **8.32** | **0.844** | **0.834** | **3.08** | **27.44** |
| Kin8nm | EDL | 0.500 | 1.07 | -0.082 | -0.117 | 0.89 | 1.26 |
| | MC Dropout | 0.722 | **0.31** | 0.260 | 0.218 | **0.17** | **0.40** |
| | Ensemble | 0.556 | 0.68 | 0.217 | 0.226 | 0.36 | 0.86 |
| | HCM | **0.842** | 0.43 | **0.642** | **0.525** | 0.18 | 0.68 |
| Power Plant | EDL | 0.444 | **39.94** | -0.050 | -0.067 | 31.36 | **48.57** |
| | MC Dropout | 0.929 | 102.64 | 0.650 | 0.651 | 38.80 | 214.07 |
| | Ensemble | 0.526 | 60.47 | 0.481 | 0.711 | 15.40 | 119.51 |
| | HCM | **1.000** | 57.82 | **0.838** | **0.969** | **7.18** | 387.25 |
| Wine Quality | EDL | 0.563 | 2.50 | 0.218 | 0.181 | 1.58 | 3.83 |
| | MC Dropout | **0.750** | **1.45** | 0.672 | 0.621 | **0.50** | **2.47** |
| | Ensemble | 0.667 | 2.48 | 0.299 | 0.288 | 1.41 | 3.73 |
| | HCM | 0.737 | 2.17 | **0.734** | **0.779** | 0.73 | 5.36 |
| Yacht | EDL | 0.750 | 0.51 | 0.701 | **0.908** | 0.11 | 9.55 |
| | MC Dropout | 0.615 | 1.38 | 0.478 | 0.428 | 0.56 | **2.02** |
| | Ensemble | 0.500 | **0.09** | 0.086 | 0.175 | **0.04** | 2.38 |
| | HCM | **0.867** | 1.48 | **0.879** | 0.855 | 0.39 | 13.91 |

Table 11: Validation MAE across $\lambda \in \{0, 1, 3, 5\}$ on six UCI regression datasets.

| Dataset | $\lambda = 0$ | $\lambda = 1$ | $\lambda = 3$ | $\lambda = 5$ |
|---------|---------------|---------------|---------------|---------------|
| Wine | 0.5539 | 0.5502 | 0.5522 | 0.5538 |
| Energy | 1.8183 | 2.7145 | 2.3795 | 2.0584 |
| Yacht | 0.0527 | 0.0536 | 0.0479 | 0.0601 |
| Concrete | 4.3095 | 4.3030 | 4.5332 | 4.6356 |
| Kin8nm | 0.0658 | 0.0641 | 0.0709 | **1.3765** |
| Power Plant | 3.4741 | 4.4440 | 5.1403 | 4.5781 |

$\lambda = 5$, where the validation MAE becomes significantly higher than for the other settings. Inspecting the training loss reveals that, under $\lambda = 5$, the constraint term overwhelms the optimization and prevents the magnitude branch $R$ from learning properly, which explains the abrupt increase in error.

Overall, the results suggest that $\lambda$ should not be viewed as a sensitive hyperparameter. In most cases, $\lambda = 0$ produces the best or one of the best results, and we therefore recommend starting from $\lambda = 0$ and increasing it gradually only if necessary. Larger values of $\lambda$ may lead to instability, especially in datasets where the unit-norm constraint becomes overly restrictive.

Table 12: AUROC for OOD detection on text classification using AG News as the in-distribution dataset. Lower values are better. Best values are in **bold**, second best are underlined.

| Method | Yelp | IMDB | Emotion | DBPedia |
|--------|------|------|---------|---------|
| HCM | 0.9229 | 0.9339 | 0.9716 | 0.9184 |
| MSP | 0.9019 | 0.8748 | 0.9720 | 0.8813 |
| HUQ | 0.9343 | 0.9235 | **0.9833** | 0.9353 |
| NUQ | **0.9695** | **0.9720** | 0.9708 | **0.9686** |
| RAU | 0.9029 | 0.8786 | 0.9453 | 0.8229 |

Table 13: FPR@95TPR for OOD detection on text classification using AG News as the in-distribution dataset. Lower values are better. Best values are in **bold**, second best are underlined.

| Method | Yelp | IMDB | Emotion | DBPedia |
|--------|------|------|---------|---------|
| HCM | 0.3388 | 0.3129 | 0.1297 | 0.3309 |
| MSP | 0.3262 | 0.3460 | 0.1369 | 0.4374 |
| HUQ | 0.1683 | 0.1636 | **0.0839** | 0.1620 |
| NUQ | **0.0598** | **0.0523** | 0.0532 | **0.0540** |
| RAU | 0.3297 | 0.3532 | 0.2588 | 0.6309 |

## A.9 ADDITIONAL OOD DETECTION RESULTS ON TEXT CLASSIFICATION

To examine the applicability of HCM beyond the domains considered in the main paper, we include supplementary OOD detection experiments on text classification. Using AG News as the in-distribution dataset and several widely used NLP corpora as OOD sources, we evaluate whether the hyperspherical decomposition consistently produces meaningful uncertainty scores in language tasks.

Across four OOD datasets, HCM shows consistently competitive AUROC performance relative to standard confidence baselines such as MSP, and clearly improves over RAU (Gong et al., 2022). Although similarity-based methods (HUQ (Vazhentsev et al., 2023) and NUQ (Kotelevskii et al., 2022)) achieve the strongest AUROC and FPR@95TPR scores—as expected from their density- and prototype-driven formulations—HCM remains close in performance despite requiring no sampling, no density estimation, and only a lightweight modification to the classifier head. Notably, HCM outperforms MSP on all datasets in AUROC, indicating that the hyperspherical deviation signal captures meaningful semantic mismatch even in the language domain. The performance on *Emotion* further suggests that HCM is robust under strong semantic shift. Overall, these results confirm that the proposed decomposition generalizes naturally to text classification without architectural tuning or task-specific adjustments.

## A.10 ADDITIONAL 1D REGRESSION

To illustrate how hyperspherical decomposition behaves in the scalar regression setting, we evaluate HCM on four controlled 1D tasks: heteroscedastic Gaussian noise, heteroscedastic Laplace noise, bimodal Gaussian mixture noise, and multi-valued regression ($y = \pm\sqrt{x}$). The same decomposition is applied by embedding each scalar target as $(y, y)$, enabling magnitude–direction factorization.

Figure 7 summarizes the 1D regression results across four noise settings. Across all noise types, HCM consistently reflects both the local noise structure and the global ambiguity of the regression task. Under Gaussian and Laplace heteroscedastic noise, HCM increases its uncertainty proportionally to the noise magnitude, successfully capturing aleatoric variability. In the bimodal setting, uncertainty grows with the separation between mixture components, and in the multi-valued task ($y = \pm\sqrt{x}$), HCM assigns larger uncertainty as the two solution branches move farther apart, reflecting the increasing discrepancy between the possible targets. Mixture Density Networks (MDN) capture aleatoric noise to a reasonable extent but do not express epistemic uncertainty. Bayesian Neural Networks (BNN) partially capture aleatoric variation but show weak epistemic sensitivity

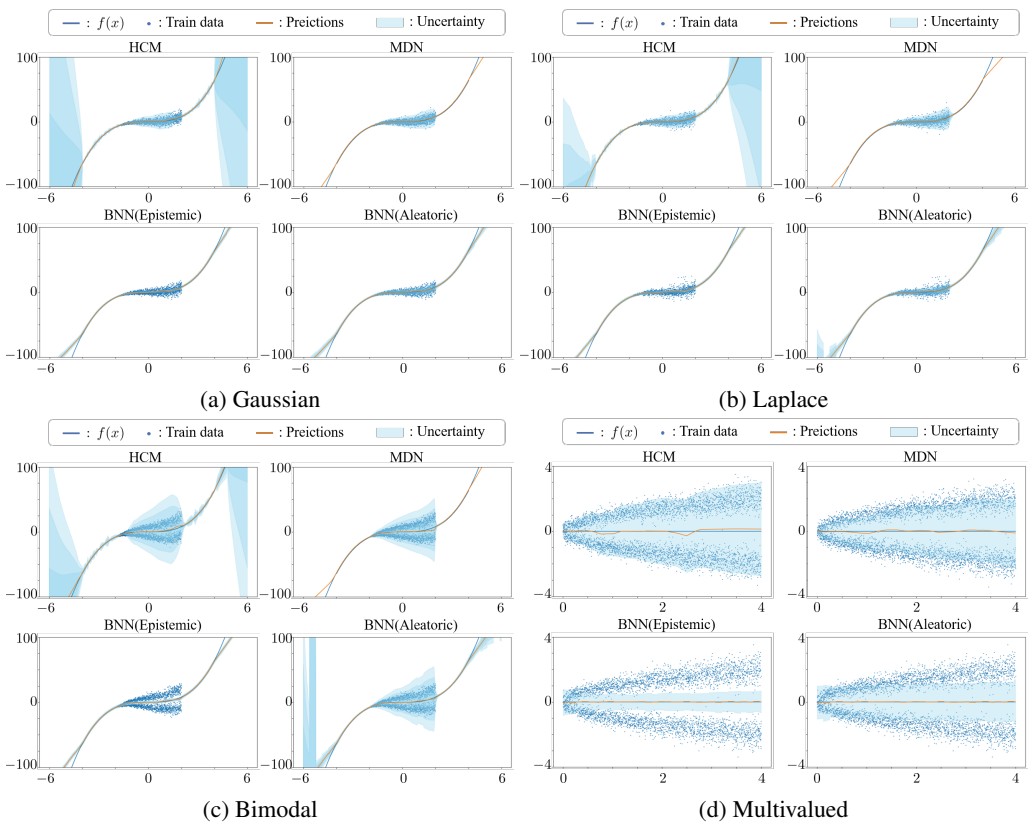

Figure 7: Additional 1D regression results under four noise structures (Gaussian, Laplace, bimodal, and multi-valued). Each panel compares HCM with MDN, BNN–epistemic, and BNN–aleatoric.

and overall unstable uncertainty estimates, which is consistent with the known difficulty of obtaining reliable uncertainty from single-network Bayesian approximations.

Overall, HCM is the only method among the compared baselines that is simultaneously sensitive to aleatoric noise and epistemic uncertainty across all noise regimes.

USE OF LARGE LANGUAGE MODELS (LLMS)

In preparing this work, we made limited use of a Large Language Model (ChatGPT, OpenAI GPT-5). Specifically:

- **Writing Assistance:** The LLM was used to improve clarity, grammar, and conciseness of English text. Draft paragraphs written by the authors were revised with its help.
- **Literature Support:** The LLM was occasionally used to retrieve references and to generate candidate BibTEX entries, which were subsequently verified by the authors.
- **Idea Organization:** The LLM assisted in structuring some sections of the paper (e.g., experimental description, figure captions), but all research ideas, methods, and conclusions were conceived and validated by the authors.

The authors have carefully reviewed, verified, and take full responsibility for all contents of the paper, including any parts drafted with the assistance of the LLM.

