# OpenReview forum: "Uncertainty Estimation via Hyperspherical Confidence Mapping"
_ICLR.cc/2026/Conference — ICLR 2026 Poster_

### Official Review · Reviewer_mKdz · 2025-10-25

**Soundness:** 3
**Presentation:** 3
**Contribution:** 3
**Rating:** 6
**Confidence:** 4

**Summary:**

In the paper, uncertainty in classification and regression is handled by transforming the target into a D-dimensional hypersphere where deviation from the norm-1 hypersphere indicates uncertainty. D is the number of classes for multi-class classification, however the way the hyperspherical target is applied to 1-dimensional regression is unclear to me, I'd like to see more details for this case.

Theoretical justification and experimentation is convincing. Unfortunately, no source code is provided.

**Strengths:**

+ Theoretically explained transformation of the learning process to include prediction uncertainty
+ Detailed experimentation
+ Well written paper

**Weaknesses:**

- the way the hyperspherical target is applied to 1-dimensional regression is unclear to me, but I may have overlooked something
- no source code is provided
- regression baselines might be extended

**Questions:**

Most importantly, the way the hyperspherical target is applied to 1-dimensional regression is unclear to me, I'd like to see more details for this case.

Pls provide source code, e.g. https://anonymous.4open.science/

The aleatoric estimator in Proposition 2 is only addressed in the case the aleatoric uncertainty is Gaussian noise. Can the proposed method handle non-Gaussian, non-unimodal, or cases when there are multiple correct answers, e.g. y=x^2(+noise)?

I would like to see qualitative or quantitative comparison with approaches such as Bayesian NN [Shengyang Sun, Changyou Chen, and Lawrence Carin. Learning Structured Weight Uncertainty in Bayesian Neural Networks. AISTATS'17], or Mixture Density Networks [Yousef El-Laham, Niccolo Dalmasso, Elizabeth Fons, and Svitlana Vyetrenko. Deep gaussian mixture ensembles. In Uncertainty in Artificial Intelligence, PMLR, 2023.].

Minor: broken reference L299 We train ResNet-18 (?)

---

> ### Author Response · Authors · 2025-11-21
>
> We thank the reviewer for the constructive feedback and for highlighting the clarity of the theoretical explanation and experiments.
> Below we address all concerns in detail.
>
> **1. Application of hyperspherical target to 1D regression**
> We appreciate the request for clarification.
> For scalar regression y∈R, we embed the target into a 2D space as (y,y), which enables the same decomposition y=Rd used in higher-dimensional regression. This preserves the symmetry required by the hyperspherical formulation.  We have now explicitly clarified this construction in Section 2.2 of the revised manuscript, and a runnable example is included in the released toy experiment code (see link below).
>
>
> **2. Source code availability**
> We now provide complete, anonymous source code for both CIFAR-10 and the 1D regression toy examples:
> https://anonymous.4open.science/r/HCM-2532/
> Additional experiment scripts will be continuously added as the remaining components are cleaned and organized.
>
> **3. Handling non-Gaussian, non-unimodal, or multi-valued noise**
> Although Proposition 2 specifically analyzes Gaussian aleatoric noise, the HCM uncertainty score itself does not assume Gaussianity.
> It depends only on constraint violation $|\||\hat{d}\||_2−1∣$, which reflects deviations arising from any form of uncertainty—Gaussian, Laplace, heavy-tailed, multimodal, or structurally ambiguous.
> We additionally provide four synthetic 1D regression studies—Gaussian heteroscedastic noise, Laplace noise, bimodal Gaussian mixture, and multi-valued regression—demonstrating that HCM behaves as expected across diverse noise structures. These results have been added to the appendix.
>
> **4. Comparison against Bayesian NN and Mixture Density Networks**
> Following the reviewer’s suggestion, we have added in the Appendix direct comparisons with:
> BNN (Bayesian Neural Networks, Shengyang Sun et al., AISTATS’17)
> MDN (Mixture Density Networks, El-Laham et al., UAI 2023)
> All models were trained under matched architecture and optimizer settings.
> Key findings (now included in the appendix):
> HCM is the only method that is sensitive to both random and epistemic uncertainty in any noise structure.
> MDN reliably captures aleatoric variability but does not express epistemic uncertainty.
> BNN learns a global aleatoric parameter and provides weak epistemic separation in these settings.
>
> **5. Minor comments**
> Broken reference (L299) has been corrected.
>
> We thank the reviewer again for the helpful comments.
> We believe the clarified 1D formulation, the newly added non-Gaussian and multimodal experiments, and the inclusion of BNN/MDN comparisons significantly strengthen the contribution.

---

> > ### Author Response · Authors · 2025-11-28
> >
> > Thank you again for the helpful review and for taking the time to evaluate our work.
> > As a brief update, we have added the full two_moons experiment code to the anonymous repository.
> > The repository (below) now includes:
> > - CIFAR-10 experiments
> > - 1D regression toy example
> > - Two moons experiments (newly added)
> >
> > https://anonymous.4open.science/r/HCM-2532/
> >
> > We hope this addition may help clarify some of the points raised, particularly regarding the behavior of the uncertainty score in low-dimensional settings. We would also like to mention that the depth estimation experiment code will be added shortly once minor refactoring is completed. For the industrial regression experiments, the code is undergoing an internal security review due to data confidentiality, and its availability may be restrictions.
> >
> > Please let us know if any further clarification would be helpful. We sincerely appreciate the reviewer’s thoughtful feedback.

---

### Official Review · Reviewer_caDX · 2025-10-27

**Soundness:** 3
**Presentation:** 3
**Contribution:** 2
**Rating:** 2
**Confidence:** 4

**Summary:**

The paper introduces an uncertainty quantification method based on decomposing model outputs into magnitude and direction components. During training, a unity-norm constraint is imposed on the direction component, and deviations from this constraint are penalized. These deviations later serve as the basis for uncertainty estimation: violations of the constraint are interpreted as indicators of uncertainty. The authors also show that prediction error can be lower-bounded by a function of this uncertainty measure. The method is evaluated on several regression and classification tasks, achieving empirical performance that is comparable to, and in some cases better than, established uncertainty quantification approaches.

**Strengths:**

One of the main strengths of the paper is the simplicity and elegance of the proposed method. The idea of using deviations from a constraint as a proxy for uncertainty is both intuitive and conceptually appealing. This design makes the approach computationally efficient while still providing a meaningful signal about the reliability of model outputs.

The empirical results further reinforce the value of the method. Performance is on par with other established uncertainty quantification techniques, and the observed correlation between the proposed uncertainty scores and prediction error suggests that the method captures relevant aspects of uncertainty effectively.

The paper is also generally well written and easy to follow, which helps convey the technical content clearly.

**Weaknesses:**

The theoretical motivation for the proposed method appears insufficiently developed. Most uncertainty quantification approaches are grounded in first principles, typically through statistical, Bayesian, or information-theoretic frameworks. In contrast, the paper lacks a similarly rigorous foundation. While the experiments demonstrate a correlation between prediction error and uncertainty scores, this empirical evidence may not be sufficient to justify the method on its own.

The paper attempts to motivate the approach through an analysis of aleatoric and epistemic uncertainty, but this connection is not convincing. The link to epistemic uncertainty, in particular, seems to rely primarily on empirical observations rather than theoretical grounding. Moreover, the proposed method does not enable a clear distinction between these two types of uncertainty. This is not a problem per se, since other established approaches such as conformal prediction also do not offer such separation, but it raises the question of why this conceptual framing is emphasized at all.

Related to these concerns, several aspects of the method remain unclear, as reflected in the questions below. These points suggest that the theoretical framing and practical interpretation could be articulated more clearly to strengthen the overall contribution.

### Minor issues
- In Remark 1, the authors note that "$u(x)$ serves as a conservative and interpretable indicator of uncertainty: it
may underestimate error, but does not overestimate it in this regime". I am not sure "conservative" is the best wording here, since underestimating the error is not typically what we associate with conservativeness. That is, I would call the method "conservative" it if gave an upper bound to the error instead.
- Res-Net 18 rference missing in line 299.

**Questions:**

1. **Dependence on Magnitude** I might be missing something, but why is the proposed uncertainty measure tied to the output magnitude? For example, in regression, could we arbitrarily shrink the uncertainty measure by scaling down the output space? Moreover, consider the following scenario where we have a 2D regression target with the ground-truth forming a circle around the origin. Would the proposed method assign low uncertainty to points close to the origin simply because the magnitude is low, even though these points are out of distribution?
2. **Interpretability of the Uncertainty Score** Could you elaborate on the interpretability of the method? I agree the constraint on the output space is intuitive and easy to follow, but the uncertainty score itself does not seem meaningful in isolation. What is its scale and unit? How does it relate to established notions of uncertainty? In Section 3.3, confidence values are derived by exponentiating the uncertainty score, but the rationale for this transformation is unclear.
3. **Dependence on Training Dynamics** In the conclusion it is said that “the uncertainty score $u(x)$ may depend on training dynamics”. Could you expand on this point? Specifically, the regularization coefficient $\lambda_{norm}$ seems likely to play a key role in the method’s effectiveness. Have you conducted any ablation studies on this parameter, or can you provide guidelines for readers on how to set it?

---

> ### Author Response · Authors · 2025-11-21
>
> We sincerely thank the reviewer for their insightful and constructive comments.
> Below, we clarify the theoretical motivation, interpretation, and practical guidance of the proposed method.
>
> **1. Theoretical motivation of the proposed uncertainty score**
> We thank the reviewer for raising this question. Our method does have theoretical grounding: Proposition 1 provides an explicit inequality showing that the prediction error $e_y$ cannot be small when the uncertainty score $u(x)$ is large. Formally, we establish a lower bound of the form $e_y\geq u(x)(1−\epsilon)$, where $\epsilon$ is a small residual term that vanishes when the model generalizes well.
> Thus, large values of $u(x)$ make low prediction error mathematically impossible—providing a necessary-condition guarantee for accuracy. The uncertainty score is therefore not only empirically correlated with error but also theoretically justified as an indicator of when accurate prediction cannot be achieved.
>
> **2. Clarification on aleatoric/epistemic framing**
> We appreciate the reviewer’s perspective. We agree that emphasizing this separation may be misleading. In the revised manuscript, we have reduced this emphasis and presented the discussion only as an intuitive interpretation, rather than a central theoretical claim.
>
> **3. Interpretability**
> We thank the reviewer for raising the question regarding the scale and units of our uncertainty score.
> Our uncertainty measure is not an arbitrary quantity: it is analytically linked to the prediction error through a provable lower bound. Although the score is not numerically equal to $\||\hat{y}−y\||_2$, it shares a consistent scale because it quantifies how far the model’s output deviates from the idealized prediction geometry. As this deviation increases, the minimum achievable error necessarily increases as well, providing a clear and interpretable relationship between the uncertainty score and prediction quality.
> Thus, larger uncertainty values correspond to a larger guaranteed lower bound on the prediction error, providing a clear and interpretable relationship between the uncertainty score and prediction quality.
>
> **4. Magnitude dependence & circle example**
> We appreciate this helpful scenario. To clarify how our method behaves when the ground-truth magnitude is small, two different scenarios must be distinguished:
> - Predicted magnitude is small (correct reason).
> In this case, the model correctly recognizes that the target lies near the origin.
> Even if the predicted direction fluctuates, the absolute prediction error remains inherently limited by the small magnitude.
> Since the maximum possible error is bounded, the resulting uncertainty naturally remains low.
> - Predicted direction has small norm (incorrect reason).
> Here, the model fails to predict both the correct magnitude and a unit-norm direction.
> A direction norm far from 1 triggers a high constraint-violation penalty, resulting in high uncertainty.
> This correctly reflects that the model did not understand the small target magnitude.
> Thus, our uncertainty measure distinguishes whether a small output arises from a correct understanding of the target or from abnormal behavior of the model.
>
> **5. Training dynamics & $\lambda$ sensitivity**
> We clarify that $\lambda$ functions as a Lagrange multiplier rather than a sensitive hyperparameter. The unit-norm constraint $\||d\||_2=1$ is naturally satisfied through minimizing the loss function, meaning the constraint is effectively embedded within the loss. Because this structure is enforced internally by the optimization dynamics, the model exhibits minimal sensitivity to $\lambda$. We present the results of an ablation study on UCI dataset.
> What matters more is the learning rate, particularly in the early stage of training. If the learning rate is too large, a single update can push ‖d‖₂ excessively far from 1, causing the constraint-violation term to spike and destabilize the optimization. To avoid this issue, we employ a warm-start procedure that stabilizes the direction vector before proceeding with full training.
>
> **Table: Validation MAE across $\lambda \in$ {0, 1, 3, 5} on six UCI regression datasets**
> | Dataset | $\lambda$=0| $\lambda$=1 | $\lambda$=3 | $\lambda$=5 |
> |-------|------|------|---------|---------|
> | Wine   | 0. 5539| 0. 5502| 0. 5522| 0. 5538|
> | Energy   | 1.8183 | 2.7145 | 2.3795 | 2.0584 |
> | Yacht   | 0.0527 | 0.0536 | 0.0479 | 0.0601 |
> | Concrete   | 4.3095 | 4.3030 | 4.5332 | 4.6356 |
> | Kin8nm   | 0.0658 | 0.0641 | 0.0709 | 1.3765 |
> | Power Plant   | 3.4741 | 4.4440 | 5.1403 | 4.5781 |
>
> **6. Minor issues**
> We removed the potentially confusing use of “conservative” and added the missing ResNet-18 reference.
>
> We thank the reviewer again for the thoughtful and constructive feedback. The comments greatly helped us clarify the theoretical motivation, refine the exposition, and improve the overall quality of the manuscript.

---

> ### Comment · Reviewer_caDX · 2025-11-24
>
> Thank you for addressing my concerns. In particular, comments 1, 3 and 4 were really helpful to better understand the motivation and overall interpretation of the method. The corresponding changes to the text also improved the paper considerably. One aspect that could still be better addressed in that regard is how to communicate this uncertainty measure to an end user who might be unfamiliar with the inner-workings of the method and might have to make an informed decision based on its output. For instance, in the two moons experiment, how did you determine that $u(x) > 0.15$ corresponds to high uncertainty? I understand the relation to the prediction error, but providing more clear guidelines on how to use the metric for this type of decision making could significantly increase its adoption by the community.
>
> Regarding the training dynamics in comment 5, the sensitivity to the $\lambda$ parameter does not seem that low. For instance, for Kin8nm the MAE increases by two orders of magnitude for $\lambda=5$. In any case, I think that is an interesting discussion to be added to the paper, including how to set the learning rate as you described.

---

> > ### Author Response · Authors · 2025-11-25
> >
> > We sincerely appreciate your positive assessment of our revised submission and your thoughtful follow-up comments. We are especially grateful for your remarks regarding the importance of providing clear, practical guidance for end users who may be unfamiliar with the internal mechanics of the method. Your suggestion to clarify how decision thresholds should be chosen is highly valuable, and we thank you for highlighting this point. In response to your comments, we provide the following clarifications and additional guidelines.
> >
> > **1. Threshold Selection for the Uncertainty Measure $u(x)$**
> > Regarding the question of how the threshold such as $u(x)>0.15$ was determined in the two–moons experiment, we would like to clarify that the appropriate decision boundary naturally depends on the nature of the task. We outline two practical strategies that we believe can assist end users in applying the method in a principled and interpretable way.
> > - Tasks with a well-defined accuracy or safety tolerance.
> > In many real-world applications—particularly industrial settings—there exists an explicit error tolerance $\epsilon$, beyond which predictions are considered unreliable or unsafe. Since our uncertainty score $u(x)$ serves as a lower bound on the prediction error, any sample satisfying $u(x)>\epsilon$ can be interpreted as violating the acceptable error margin.
> > In these cases, the threshold is not arbitrary; it directly follows from the domain-specific requirements.
> >
> > - Tasks without an explicit tolerance.
> > When no predefined error criterion exists, a distributional calibration approach can be adopted. A simple and practical guideline is to compute the empirical distribution of $u(x)$ on the validation set and select a high quantile (e.g., 95% or 99%) as the threshold.
> > This identifies samples whose uncertainty is unusually large relative to typical in-distribution behavior, making them natural candidates for “high-uncertainty” regions.
> >
> > In the two–moons experiment, the threshold 0.15 was chosen following the second strategy: it corresponds approximately to the upper tail of the validation-set uncertainty distribution and therefore provides a reasonable and interpretable cutoff for identifying ambiguous or out-of-regime points. We have added a brief explanation of these thresholding guidelines to Section 2.7 of the main paper.
> >
> > **2. The sensitivity to the Regularization Parameter $\lambda$**
> >
> > Regarding your comment on the sensitivity to the $\lambda$ parameter, we appreciate you drawing attention to this point. As you noted, certain tasks—such as Kin8nm—exhibit unusually high loss values when $\lambda$ is set to a large value (e.g., $\lambda$=5). We agree that this behavior deserves further clarification for readers.
> > To better understand this phenomenon, we tracked the training dynamics of such cases and found that overly large $\lambda$ values impose excessively strong constraints on the unit-norm regularization. As a result, the loss term associated with the magnitude component $R$ fails to decrease, preventing the model from learning meaningful representations. This explains the sudden increase in MAE for tasks that are particularly sensitive to over-regularization. In our experience, small values (e.g., $\lambda\leq1$) generally work well, while larger values should be used with caution.
> > We have incorporated this discussion into the revised manuscript, both in the Limitations section and Appendix A.8.
> >
> > We sincerely thank you once again for your thoughtful and constructive comments. Your insights have significantly improved the clarity and practical value of the paper.

---

### Official Review · Reviewer_X9Js · 2025-10-31

**Soundness:** 3
**Presentation:** 3
**Contribution:** 2
**Rating:** 4
**Confidence:** 3

**Summary:**

The paper introduces a new sampling-free uncertainty quantification framework, termed Hyperspherical Confidence Mapping (HCM). The authors establish the theoretical foundations of this approach by reformulating the task as a constrained optimization problem related to prediction error. The proposed framework effectively disentangles uncertainty into its aleatoric and epistemic components. Comprehensive evaluations on both classification and regression image datasets demonstrate the superior performance of HCM in academic benchmarks and real-world semiconductor manufacturing tasks, underscoring its practical applicability.

**Strengths:**

1. The paper is well-structured and clearly presents the underlying motivation and experimental results.
2. The authors provide a solid theoretical foundation for the proposed method, enabling the disentanglement of uncertainty into two components.
3. The experimental evaluation demonstrates superior performance across several image classification and regression tasks, including both public datasets and industrial applications.

**Weaknesses:**

1. The core idea of hyperspherical mapping, while interesting and effective, is not entirely novel. Previous work has already explored a similar concept in the context of text classification [1]. Although there are differences between the two approaches, these distinctions should be clearly articulated and supported through comparative analysis in the experimental setup.
2. Table 1 suggests that similarity-based methods are not interpretable. However, the decisions made by such methods can, in fact, be interpreted, for example, through dimensionality reduction techniques applied to their hidden representations.
3. Some relevant works from the similarity-based group are missing and should be included in the experimental evaluation, such as NUQ [2] and HUQ [3].
4. The experimental evaluation could be strengthened by extending the approach to text classification tasks [1,2,3].
5. Broken reference on line 299.

[1] Gong et al. Confidence Calibration for Intent Detection via Hyperspherical Space and Rebalanced Accuracy-Uncertainty Loss. AAAI 2022.\
[2] Kotelevskii et al. Nonparametric Uncertainty Quantification for Single Deterministic Neural Network. NeurIPS 2022. \
[3] Vazhentsev et al. Hybrid Uncertainty Quantification for Selective Text Classification in Ambiguous Tasks. ACL 2023

**Questions:**

1. Are there any specific considerations or challenges when applying this approach to other domains?

---

> ### Author Response · Authors · 2025-11-21
>
> We sincerely thank the reviewer for the constructive and detailed feedback.
> Below, we address each point and clarify how the revised manuscript incorporates the reviewer’s suggestions.
>
> **1. On the novelty of hyperspherical mapping for uncertainty estimation**
> We appreciate the reviewer’s observation regarding prior work that applies hyperspherical representations to text classification (Gong et al., AAAI 2022).
> While the surface-level idea of using a hyperspherical target appears similar, the underlying formulation and objectives differ fundamentally from our approach:
> Prior work uses hyperspherical classification logits primarily to improve intention detection and calibration.
> In contrast, our method decomposes the regression/classification target itself into a magnitude–direction pair and interprets constraint violations as uncertainty . This leads to a sampling-free, theoretically motivated uncertainty score with a closed-form lower bound on prediction error, which is not present in prior hyperspherical text methods.
> Moreover, unlike similarity/distance-based methods, HCM does not rely on class prototypes or density estimation, enabling applicability to regression tasks as well.
> We have clarified these distinctions in Section 2.2 and added explicit discussion comparing HCM with hyperspherical calibration approaches.
>
> **2. Additional text classification experiments**
> The reviewer suggested evaluating HCM on text classification tasks.
> We conducted new experiments using AG News (ID) with four OOD datasets: Yelp, IMDB, Emotion, DBPedia, following a BERT-base encoder architectural changes. Specifically, we added a lightweight head that decomposes the final hidden representation into a magnitude component R and a unit-norm direction componentd, while keeping all BERT parameters frozen. We include the full experimental setup and results in Appendix.
> The results show that:
>
> **Table: AUROC for OOD detection on text classification (AG News as ID).
> Higher values are better. Best values are in bold\**
>
> | Method | Yelp | IMDB | Emotion | DBPedia |
> |-------|------|------|---------|---------|
> | HCM   | 0.9229 | 0.9339 | 0.9716 | 0.9184 |
> | MSP   | 0.9019 | 0.8748 | 0.9720 | 0.8813 |
> | HUQ   | 0.9343 | 0.9235 | **0.9833** | 0.9353 |
> | NUQ   | **0.9695** | **0.9720** | 0.9708 | **0.9686** |
> | RAU   | 0.9029 | 0.8786 | 0.9453 | 0.8229 |
>
> HCM consistently outperforms MSP in AUROC across all OOD benchmarks.
> HCM is competitive with HUQ/NUQ despite requiring no density estimation, sampling, or additional classifiers.
> HCM generalizes to NLP tasks without any architectural tuning or task-specific adaptations.
> These findings confirm that the hyperspherical decomposition naturally extends to the text domain.
>
> **3. On similarity-based methods and interpretability**
> We agree with the reviewer that similarity-based methods can be interpreted through feature-space visualization or clustering techniques.
> By **interpretability**, we intended to highlight the contrast between:
> methods whose primary mechanism is similarity-based scoring (e.g., energy, distance, HUQ/NUQ) that rely on feature-space interpretation, and our method, whose uncertainty arises from constraint violations in the target-space through hyperspherical decomposition. To avoid misunderstanding, we have revised Table 1 to acknowledge that similarity-based approaches can indeed be interpreted, even if their interpretability requires additional embedding-space analysis.
>
> **4. Missing baselines (NUQ, HUQ).**
> We thank the reviewer for pointing out the absence of NUQ and HUQ in our evaluation.
> We did not include these methods because they are not included in the standard OpenOOD protocol we followed. However, in response to the reviewer’s suggestion, we have now conducted additional experiments on text classification, where NUQ and HUQ are more commonly applied, and reported the full results in Appendix.
> This supplementary evaluation confirms that HCM remains competitive with similarity-based uncertainty methods even outside the vision domain.
>
> **5. Broken reference**
> The broken reference at line 299 has been fixed.
>
> **6. Broader applicability to other domains.**
> In response to the reviewer’s question, HCM does not require any assumptions about the input domain.
> Since the hyperspherical formulation modifies only the structure of the **output** vector, the method applies to any modality—vision, text, tabular, or spectral— so long as the model can reliably produce meaningful output representations.
> The only practical limitation arises when the target space itself cannot be represented in a magnitude–direction form (e.g., inherently multi-valued or set-based outputs).
> We have clarified this point and included the corresponding discussion in the Limitations section in the Conclusion.
>
> We appreciate the reviewer’s thoughtful comments, which helped us improve the clarity and completeness of the paper.

---

> > ### Comment · Reviewer_X9Js · 2025-11-26
> >
> > Thank you for the thoughtful and detailed response. The authors have addressed most of the raised concerns, which improves the clarity and strength of the paper. Accordingly, I slightly increase my rating.

---

> > > ### Author Response · Authors · 2025-11-26
> > >
> > > We would like to express our sincere gratitude for your thoughtful follow-up and for taking the time to reassess our submission. We truly appreciate the care with which you reviewed our rebuttal. Your detailed comments greatly helped us refine the presentation, clarify the novelty, and strengthen the experimental discussion.
> > >
> > > We are grateful that the revisions addressed most of your concerns, and we sincerely appreciate your decision to raise the rating. Thank you again for your constructive, fair, and encouraging feedback throughout the review process.

---

### Official Review · Reviewer_MWzw · 2025-11-02

**Soundness:** 3
**Presentation:** 3
**Contribution:** 3
**Rating:** 6
**Confidence:** 3

**Summary:**

The paper proposes a novel approach for uncertainty estimation (UE), Hyperspherical Confidence Mapping (HCM), based on the decomposition of the output of the model on the magnitude scalar value (R) and unit-norm direction vector (d). To apply this approach, authors modify the models to predict both R and d values and decompose the original classification or regression target into two corresponding components. After that, authors define a composite loss function for training models with aforementioned changes, and quantify an uncertainty score u(x) based on predicted R and d values. Moreover, the authors also separate epistemic and aleatoric uncertainties and define a separate estimator for aleatoric uncertainty, while u(x) serves as the epistemic uncertainty score. Finally, the authors conducted thorough experiments on OOD detection on image classification, uncertainty calibration for regression for depth estimation, and uncertainty calibration for an industrial regression task. All of the experiments show the applicability of the HCM and demonstrate that HCM shows comparable performance with the other UE methods.

**Strengths:**

1.	The proposed HCM approach provides a novel view on uncertainty estimation and provides theoretical foundations and interpretation for the HCM UE score.
2.	The experiments on Two Moons demonstrate the interpretability of the method, while experiments on calibration and OOD detection show that the HCM achieves comparable performance with other UE methods or outperforms them.
3.	The experiments on industrial data further strengthen the claim about HCM’s applicability and demonstrate that HCM outperforms other methods on real-world data for the calibration task.

**Weaknesses:**

1.	HCM requires additional model fine-tuning. This can limit the method’s applicability for some modern models (e. g. in a zero-shot setting). Moreover, if the presented HCM approach is used for obtaining predictions on the main task – for example, one can use R and d value to both obtain predicted class and to estimate uncertainty – it can affect the performance of the model on the main task. This topic was left unexplored during the experiments; however, it can significantly affect the applicability of the HCM.
2.	Limited applicability on the OOD detection task. On the OOD detection task, HCM shows comparable performance with a much simpler MDS method, which does not require model modification – these limit the HCM applicability for OOD detection, especially in the areas where fine-tuning is rarely used (e. g. LLMs).

**Questions:**

1.	Missed reference on line 299.
2.	Typo in line 749: “gound” -> “ground”.

---

> ### Author Response · Authors · 2025-11-21
>
> We thank the reviewer for the thoughtful and constructive feedback. We greatly appreciate the time and care taken to assess our work.
>
> **1. Fine-tuning requirement and zero-shot applicability**
> We appreciate the reviewer’s comment regarding HCM’s need for model modification and light fine-tuning. We agree that this requirement can limit applicability in strict zero-shot scenarios, and we have explicitly added this point to the Limitations section. At the same time, we note that many practical UE applications—such as our industrial setting—naturally rely on fine-tuned or supervised models, where HCM can be applied without difficulty. Extending HCM to frozen-encoder or zero-shot models is an interesting future direction, and we now discuss this possibility in the revised manuscript.
>
> **2. Potential performance degradation due to decomposition into $R$ and $d$**
> We also acknowledge the reviewer’s concern that introducing the magnitude–direction decomposition may cause slight accuracy reductions, for example through the multiplicative interaction between R and d or the unit-norm constraint on d. We agree that such effects can appear, and we have clarified this behavior in Section 3.3. Across Table 3, Table 4, and Appendix Table 10, a small accuracy drop is indeed observed in a few tasks. However, the drop is generally minor and does not outweigh the benefits gained from improved uncertainty characterization. We now highlight this trade-off in Section 3.3 to make the implications transparent to readers.
>
> **3. Minor issues**
> We have fixed the typo (“gound” → “ground”) and added the missing reference noted by the reviewer.
>
> We thank the reviewer for the constructive feedback, which helped us improve both clarity and completeness.

---

### Author Response · Authors · 2025-11-29

Dear Area Chair,

Thank you for taking on the considerable responsibility of evaluating submissions in such an unexpected and challenging review cycle. We sincerely appreciate the additional effort required to ensure a fair and thorough assessment given the recent incident.

To assist your evaluation, we summarize below the revisions made in direct response to each reviewer’s comments.

**[Reviewer caDX: Theoretical Motivation & Interpretation]**
We thank Reviewer caDX for raising important questions regarding the theoretical foundation, interpretability, and magnitude behavior of our uncertainty score. In response:
- We strengthened the theoretical justification through Proposition 1, clarifying how the uncertainty score provides a lower bound on prediction error.
- The geometric interpretation underlying the magnitude–direction decomposition has been expanded.
- We reduced potential confusion regarding aleatoric vs. epistemic components.
- We addressed edge cases related to magnitude behavior.

These revisions were incorporated into the manuscript, and the reviewer noted that the updated explanations significantly improved his/her understanding.

**[Reviewer X9Js: Applicability, Novelty & Comparisons]**
We appreciate Reviewer X9Js for highlighting the importance of contextualizing HCM relative to hyperspherical text methods and similarity-based baselines. In response:
- We added a detailed comparison in Section 2.2, clarifying conceptual distinctions between HCM and prior hyperspherical approaches.
- We expanded the evaluation with a text classification/OOD detection suite (AG News + four OOD datasets).
- We added NUQ and HUQ as additional competitive baselines.
- We revised Table 1 to avoid overstating limitations of similarity-based methods.

The reviewer acknowledged that these updates addressed the majority of his/her concerns.

**[Reviewer mKdz: 1D Regression, Code Release & Baselines]**
We thank Reviewer mKdz for requesting clarity on how HCM applies to 1D regression. Accordingly:
- We added a detailed explanation of the 1D hyperspherical target in Section 2.2.
- We released complete anonymized source code, covering CIFAR-10, two-moons, and 1D regression.
- We added four additional non-Gaussian and multimodal regression tasks, along with comparisons to BNN and MDN.

After these additions, the reviewer raised no further concerns.

**[Reviewer MWzw: Fine-Tuning Requirements & Performance Trade-offs]**
We appreciate Reviewer MWzw for pointing out practical considerations regarding model modifications:
- We clarified in the Limitations section that HCM requires light model adaptation, noting that while this constrains strict zero-shot usage, it aligns well with fine-tuned or supervised settings.
- We further explained potential minor accuracy drops arising from magnitude–direction decomposition, highlighting this trade-off in Section 3.3.

No additional issues were raised following these clarifications.

We sincerely appreciate the time, care, and thoughtful evaluation you have provided during this challenging review cycle. All reviewer-requested revisions have been incorporated into the manuscript, with changes marked in red for easy reference. Please let us know if we can provide any further information. Thank you again for your dedication and fairness throughout this process.

Authors

---

### Meta-Review · Area_Chair_o7uc · 2025-12-20

**Summary:**

Overall, the reviewers found the work to be interesting and to contain novel ideas. However, several concerns were raised regarding the theoretical justification, motivation, and interpretation of the proposed method; its application in regression settings; its novelty in light of prior work; the inclusion of code to reproduce the experiments; and the need for additional numerical results. In their rebuttal, the authors have managed to address most, if not all, of these concerns and have incorporated the corresponding revisions into the paper.

**Reviewer Concerns:**

I believe most, if not all, concerns by the reviewers, as mentioned above, have been addressed by the rebuttal.

**Reviewer Scores:**

Two reviewers (X9Js and caDX), who initially gave the lowest ratings, explicitly indicated that they had either increased their scores after the rebuttal or that their concerns had been addressed, with only minor follow-up questions remaining. In light of this, I believe that most reviewers would now rate the paper as marginally above the acceptance threshold.

---

### Decision · Program_Chairs · 2026-01-26

Accept (Poster)